# Compositional Generalization from Learned Skills via CoT Training: A Theoretical and Structural Analysis for Reasoning

**Xinhao Yao[1,2,3], Ruifeng Ren[1], Yun Liao[4], Lizhong Ding[5], Yong Liu[1,2,3]***

[1]Gaoling School of Artificial Intelligence, Renmin University of China, Beijing, China
[2]Beijing Key Laboratory of Research on Large Models and Intelligent Governance
[3]Engineering Research Center of Next-Generation Intelligent Search and Recommendation, MOE
[4]Tianjin University of Science and Technology, [5]Beijing Institute of Technology
{yaoxinhao021978, liuyonggsai}@ruc.edu.cn

## Abstract

Chain-of-Thought (CoT) training has markedly advanced the reasoning capabilities of large language models (LLMs), yet the mechanisms by which CoT training enhances generalization remain inadequately understood. In this work, we demonstrate that compositional generalization is fundamental: models systematically combine simpler learned skills during CoT training to address novel and more complex problems. Through a theoretical and structural analysis, we formalize this process: ❶ Theoretically, the information-theoretic generalization bounds through distributional divergence can be decomposed into in-distribution (ID) and out-of-distribution (OOD) components. Specifically, the non-CoT models fail on OOD tasks due to unseen compositional patterns, whereas CoT-trained models achieve strong generalization by composing previously learned skills. In addition, controlled experiments and real-world validation confirm that CoT training accelerates convergence and enhances generalization from ID to both ID and OOD scenarios while maintaining robust performance even with tolerable noise. ❷ Structurally, CoT training internalizes reasoning into a two-stage compositional circuit, where the number of stages corresponds to the explicit reasoning steps during training. Notably, CoT-trained models resolve intermediate results at shallower layers compared to non-CoT counterparts, freeing up deeper layers to specialize in subsequent reasoning steps. A key insight is that CoT training teaches models how to think—by fostering compositional reasoning—rather than merely what to think, through the provision of correct answers alone. This paper offers valuable insights for designing CoT strategies to enhance LLMs' reasoning robustness[1].

## 1 Introduction

Training large language models (LLMs) to generate explicit reasoning chains marks a pivotal shift in AI development. This Chain-of-Thought (CoT) [109] paradigm moves beyond mere data scaling [39, 32] toward cultivating advanced reasoning strategies [55], resulting in more effective learning outcomes [132, 129, 105, 30, 85, 128, 43, 33, 31, 55]. Leading organizations—such as OpenAI, through reinforcement fine-tuning (RFT/ReFT) [96] of its O1 models [67], and DeepSeek, via the incorporation of long CoT cold-start data in DeepSeek-R1 [25]—are increasingly treating step-by-step reasoning annotations not merely as a prompting tool, but as a fundamental training objective.

Despite its empirical success, the underlying mechanisms of CoT remain a widely debated topic [75]. Recent studies demonstrate that CoT enhances transformers' [97] expressiveness and computational capacity, enabling them to solve problems in higher complexity classes when processing sufficiently long sequences [17, 63, 53, 73]. Other prior efforts have advanced understanding through case studies of functions learnable in-context with CoT [50, 5, 48, 116] and analyses of multi-step

---

*Corresponding author.

[1]Code is available at https://github.com/chen123CtrlS/T-CotMechanism

model estimation errors [35, 19]. However, these approaches do not explain how such capabilities emerge during the training of transformers to generate reasoning chains—precisely when core model competencies are being formed [54, 138]. In light of this concern, Kim & Suzuki [42], Wen et al. [110] focus on the parity problem and highlight that CoT improves the sample efficiency of transformers. Meanwhile, Wang et al. [106] assess different data collection strategies and point out that the granularity of CoT data strongly correlates with model performance. Crucially, it remains unclear: *(Q1)* Does CoT training improve reasoning generalization across both in-distribution (ID) and out-of-distribution (OOD) scenarios—and if so, what theoretical principles govern this ability? *(Q2)* How is this generalization capability realized within the model's internal representations?

In this paper, we propose that **compositional generalization is fundamental**: *models systematically combine simpler learned skills during CoT training to address novel and more complex problems.* We formalize this process through a unified theoretical framework and validate it via a detailed structural analysis of the model's internal computational circuits.

• **Generalization Error Analysis** (Section 2, for **Q1**). We provide an in-depth theoretical analysis to substantiate this proposition. Specifically, we first examine the KL divergence $D_{KL}(\cdot)$ between the training and test distributions. We then employ information-theoretic [114, 78, 86, 108, 107] methods to derive generalization bounds based on distributional divergence. Theorem 1 (Remark 1) shows the generalization error naturally decomposes into ID and OOD components: (i) For ID, the compositional patterns of the test problems **exactly match** those seen in the training set. The model only needs to recognize the problem type and retrieve the corresponding compositional pattern learned during training. As a result, the generalization error approaches zero under sufficient training (i.e., $D_{KL}(P_{test}^{ID}||P_{train}) = 0$), regardless of the use of CoT. (ii) For OOD, models trained without CoT fail to generalize, as the OOD test problems feature **novel** compositional patterns of learned, simpler skills. In contrast, CoT-trained models excel in these settings by leveraging exposure to the necessary simpler skills during training, achieving near-perfect OOD performance (Theorem 2). (iii) Additionally, adding noise to training data raises the generalization error upper bounds for both ID and OOD settings, and the error bound grows with the level of noise.

Our experiments serve as empirical support for the theoretical claims above. We begin with controlled data distributions to mitigate the complexity inherent in real-world training and enable precise mechanistic analysis. We demonstrate that CoT training accelerates convergence and substantially improves generalization, extending it from ID to both ID and OOD settings (Section 3.2). This confirms that CoT enables compositional generalization, whereas non-CoT models only rely on fixed input-output mappings. Next, Section 3.4 shows that CoT generalizes robustly to noisy reasoning steps within a tolerable noise level, underscoring the importance of reliable compositional generalization. Finally, validation on real-world datasets (e.g., math word problems) shows that learned skills and compositional patterns remain applicable in complex scenarios.

• **Internal Circuits of CoT vs. Non-CoT Training** (Section 3.3, for **Q2**). Using logit lens and causal tracing experiments, we find that CoT training induces a two-stage compositional circuit characterized by sparse token dependencies. This structure **reflects the process of composition**: the *model progressively internalizes the reasoning process—each explicit reasoning step during training maps onto a distinct stage within the circuit.* Conversely, Wang et al. [102] show that similar circuits emerge only during ID generalization, suggesting that without CoT, reasoning steps are not systematically internalized. Interestingly (Figure 2), CoT training allows the intermediate result to be extracted from a lower layer (e.g., index 3) for ID examples, whereas Non-CoT Training requires a higher layer (e.g., index 5). *Intuitively, a smaller layer index implies that more layers remain available for processing the second reasoning step, potentially enhancing model performance.*

**Main contributions**. Briefly, we establish a theoretical and structural framework for understanding the generalization of CoT training. A key point is that CoT training helps models **learn how to think**—by enabling compositional reasoning—**not just what to think**, by simply providing correct answers. These findings offer insights into CoT strategies for LLMs to achieve robust generalization.

# 2 THEORETICAL ANALYSIS

## 2.1 PRELIMINARIES

To begin with, we formally introduce the data distribution setting, clarify the distinctions between ID and OOD generalization in the context of **compositional reasoning tasks**, and give the definition of expected generalization error. These form the basis for our subsequent theoretical analysis.

**Definition 1** (Data Distribution). *Let $X$ and $Y$ denote the input and output random variables, respectively. The data distribution and its underlying relationships are characterized on the conditional distribution $P(Y \mid X)$. Let CoT be a sequence $C = (C_1, C_2, \cdots, C_K)$, where each $C_k$ denotes the reasoning state at the $k$-th step, then we can get the following decomposition:*

$$P(Y \mid X) = \sum_C P(Y|X,C)P(C|X) = \sum_C P(Y|X,C) \prod_{k=1}^K P(C_k|C_{<k}, X),$$

*where $P(C_k|C_{<k}, X)$ is the probability of the $k$-th reasoning step given all previous steps and the input. Each unique sequence of $C$ corresponds to a distinct compositional pattern. **ID and OOD are thus defined by whether these patterns are shared with the training data or are novel and unseen**. Representative examples are provided in Appendix C.*

Recently, information-theoretic generalization bounds [114, 78, 86, 108, 107] have been introduced to analyze the expected generalization error of learning algorithms. A key benefit of these bounds is that they depend not only on the data distribution but also on the specific algorithm, making them an ideal tool for studying the generalization behavior of models trained using particular algorithms.

**Definition 2** (Expected Generalization Error (informal)). *From a traditional statistical learning perspective, the expected generalization error is defined as the difference between the population risk and the empirical risk. Intuitively, the empirical risk reflects performance on the training data, while bounds on the generalization error allow us to estimate the population risk on unseen test data. We denote the expected generalization error by $\widetilde{error}$, see Appendix B.2 for a detailed definition.*

## 2.2 INFORMATION-THEORETIC GENERALIZATION BOUNDS

In this part, $P_{\text{train}}(Y|X)$ and $P_{\text{test}}(Y|X)$ denote the conditional distributions of the training and test data, respectively. Based on this, we provide a theoretical analysis of the divergence between the training and test distributions, along with the resulting generalization error bounds.

Since compositional patterns in ID test data (and training data) **do not appear** in OOD test data, we make a plausible assumption[2] about $P_{\text{test}}(Y|X)$:

**Assumption 1** (Mixed Test Distribution). *Assume $P_{test}(Y|X)$ is a mixture of two distributions with disjoint support: $P_{test}^{ID}(Y|X)$ and $P_{test}^{OOD}(Y|X)$. Then*

$$P_{test}(Y|X) = (1-\alpha)P_{test}^{ID}(Y|X) + \alpha P_{test}^{OOD}(Y|X),$$

*where $\alpha \in [0, 1]$ is the mixing coefficient of the OOD data.*

Consequently, the distributional divergence between the test and training sets can be assessed quantitatively, which helps us characterize the expected generalization error as follows:

**Theorem 1** (Generalization Bounds via Distributional Divergence). *Under the conditions specified in Assumption 1 and Definition 2, we assume that the loss $\ell(w, Z)$ is R-subGaussian for any $w \in \mathcal{W} \in \mathbb{R}^d$, then the expected generalization error is bounded by:*

$$\widetilde{error} \le \sqrt{\frac{2R^2}{N} \left[ (1-\alpha)D_{KL}(P_{test}^{ID}(Y|X)||P_{train}(Y|X)) + \alpha D_{KL}(P_{test}^{OOD}(Y|X)||P_{train}(Y|X)) \right]},$$

*where $N$ is the training data size, $Z = (X, Y)$ and $\mathcal{W}$ is the space of hypotheses related to the model. $\alpha$ denotes the mixing coefficient of the OOD data in test distribution and $D_{KL}(\cdot)$ represents the KL divergence between ID/OOD test and training distributions.*

---

[2]Linear combining rule is a good starting point for target mixture distribution [60, 16, 90, 58, 2].

**Remark 1** (Theoretical Analysis of ID/OOD Generalization with/without CoT). *Theorem 1 shows the generalization error naturally decomposes into ID and OOD components with coefficient $\alpha$. ❶ For ID, **the compositional patterns $C$ in the test problems exactly match those encountered during training**, since the ID test data comprises instances derived from known compositions in training data. Thus, under sufficient training—with or without CoT—the ID generalization error approaches zero (i.e., $D_{KL}(P_{test}^{ID}||P_{train}) = 0$), consistent with the results in Figure 1 (Left Part). ❷ For OOD, considering the data generation process, we have: $P(Y|X) = \sum_C P(Y|X,C)P(C|X)$, where $C$ is the CoT reasoning sequence: (a) when trained without CoT (Figure 1 (Left)), the model does not explicitly learn the distribution $P(C|X)$ (i.e., $C$ never appears during training), the $P(C|X)$ becomes a uniform prior, making $P(Y|X,C)$ reduce to $P(Y|X)$. Moreover, since OOD test problems involve unseen compositional patterns, training without CoT struggles to generalize to OOD settings. (b) In contrast, with CoT training, both $P(C|X)$ and $P(Y|X,C)$ are explicitly modeled (**that is, the learned, simpler skills**), aligning precisely with the two stages of the compositional circuit (Figure 2, Right). CoT training enables models to generalize near-perfectly to OOD examples by exposing them to simpler sub-tasks during training (Figure 1 (Center Right)). When the compositional structure aligns between training and OOD tasks, the model can effectively apply its learned reasoning skills. CoT training **teaches models how to think through compositional reasoning**—not merely what to think (matching the answers). For specifics, refer to Theorem 2.*

**Theorem 2** (OOD Generalization Error for Training with CoT). *Let the conditions specified in Theorem 1 hold, and assume[3] sufficient training such that $D_{KL}(P_{test}^{ID}||P_{train}) \to 0$. Define $P(Y|X) = \sum_C P(Y|X,C)P(C|X)$, then the OOD generalization error is bounded by:*

$$\widetilde{error}^2 \leq \frac{2R^2\alpha}{N}[D_{KL}(P_{test}^{OOD}(C|X)||P_{train}(C|X))$$
$$+ \mathbb{E}_{C \sim P_{test}^{OOD}(C|X)}[D_{KL}(P_{test}^{OOD}(Y|X,C)||P_{train}(Y|X,C))]],$$

*where $C$ denotes the CoT intermediate reasoning steps, see Appendix B.4 for more extensions.*

**Remark 2** (Robustness Discussion of Training with CoT). *For CoT training in the presence of erroneous reasoning steps, let the distribution after adding noise to $P_{train}(Y|X)$ be denoted as $P_{new}(Y|X)$. Briefly, adding noise leads to an increase in both $D_{KL}(P_{test}^{ID}||P_{new})(> D_{KL}(P_{test}^{ID}||P_{train}))$ and $D_{KL}(P_{test}^{OOD}||P_{new})(> D_{KL}(P_{test}^{OOD}||P_{train}))$. This corresponds to an increase in the upper bounds of the generalization error for both ID and OOD cases, and the larger the noise ratio $\xi$, the higher the error bounds. A more detailed theoretical explanation is provided in Theorem 3 (Appendix B.6).*

## 3    SYSTEMATIC GENERALIZATION VIA A TWO-STAGE CIRCUIT

This section further illustrates the compositional generalization, with our experiments serving as empirical support for the aforementioned theoretical claims. We first show that CoT significantly accelerates convergence and improves generalization compared to training without it (Section 3.2), then examine the internal mechanisms of CoT training, focusing on a two-stage compositional circuit (Section 3.3). Next, we analyze the robustness of CoT training in the presence of erroneous reasoning steps (Section 3.4). Finally, we validate our findings on real-world datasets (Section E).

### 3.1    GENERAL SETUP

Here, we describe the core components of our study by reviewing some basic notations. Briefly, we have the model learn all atomic facts as "simpler learned skills", while treating multi-hop facts as "complex problems". Relational patterns $(r_1, \cdots, r_n)$ are used to simulate compositional patterns.

**Atomic and Multi-Hop Facts.** The atomic (one-hop) fact describes two entities and the relationship between them, which can be represented as a triplet $(e_1, r_1, e_2)$. For example, "The United States' capital is Washington, D.C." More specifically, $e_1$ refers to the head entity (e.g., the United States), $r_1$ is the relation (e.g., Capital), and $e_2$ refers to the tail entity (e.g., Washington, D.C.). Based on atomic facts, a two-hop fact can be derived by combining two atomic facts, such as $(e_1, r_1, e_2) \oplus (e_2, r_2, e_3) \implies (e_1, r_1, r_2, e_3)$, where $e_2$ serves as a bridge entity. Similarly, a multi-hop fact can be constructed recursively from more atomic facts, resulting in $(e_1, r_1, e_2) \oplus (e_2, r_2, e_3) \oplus ... \oplus$

---

[3]This correlates to what architectures and tasks we are considering, see Figure 1 and Section 3.1 for details.

$(e_n, r_n, e_{n+1}) \implies (e_1, r_1, r_2, ..., r_n, e_{n+1})$, where $n$ is the number of steps. In real-world scenarios, $e_2$ corresponds to the intermediate inference in a two-hop fact, while $e_2, e_3, ..., e_n$ are intermediate results in more complex multi-step reasoning.

**Training Data.** Firstly, following Wang et al. [102], we define the sets of entities $\mathcal{E}$ and relations $\mathcal{R}$, from which the set of atomic facts is constructed as $\mathcal{S} = \{(e_1, r_1, e_2)|e_1, e_2 \in \mathcal{E}, r_1 \in \mathcal{R}\}$. Our training set of atomic facts, denoted as $S$, is sampled from $\mathcal{S}$ (i.e., $S \subset \mathcal{S}$). Then, we partition $S$ into two subsets, $S_{\text{ID}}$ and $S_{\text{OOD}}$ ($S_{\text{ID}} \cup S_{\text{OOD}} = S, S_{\text{ID}} \cap S_{\text{OOD}} = \emptyset$), which are used to form two sets of two-hop facts, $S_{\text{ID}}^{(2)}$ and $S_{\text{OOD}}^{(2)}$, where $S_{\text{ID}}^{(2)} = \{(e_1, r_1, r_2, e_3)|(e_1, r_1, e_2), (e_2, r_2, e_3) \in S_{\text{ID}}, e_1 \neq e_3\}$, and $S_{\text{OOD}}^{(2)}$ is formed in a similar manner. To evaluate the model's ID and OOD generalization ability, the training dataset $T$ excludes $S_{\text{ID}_{\text{test}}}^{(2)}, S_{\text{OOD}}^{(2)}$, that is, $T = S_{\text{ID}} \cup S_{\text{OOD}} \cup S_{\text{ID}_{\text{train}}}^{(2)}$, where $S_{\text{ID}_{\text{train}}}^{(2)} \cup S_{\text{ID}_{\text{test}}}^{(2)} = S_{\text{ID}}^{(2)}$. Notice that $S_{\text{ID}} \cap S_{\text{OOD}} = \emptyset$, so $S_{\text{ID}}^{(2)} \cap S_{\text{OOD}}^{(2)} = \emptyset$ and the relation compositions $(r_i, r_j)$ in $S_{\text{ID}}^{(2)}$ will not appear in $S_{\text{OOD}}^{(2)}$. Specifically, $S_{\text{ID}_{\text{train}}}^{(2)}$ and $S_{\text{ID}_{\text{test}}}^{(2)}$ are disjoint subsets whose relation compositions are sampled from the same space, enabling evaluation on unseen instances within known compositions. In contrast, $S_{\text{OOD}}^{(2)}$ involves entirely novel relation compositions, drawn from a disjoint set of relation types, thus testing generalization to unseen compositional patterns. Let $\lambda = |S_{\text{ID}_{\text{train}}}^{(2)}|/|S_{\text{ID}}|$, where |set| denotes the number of samples in the set. **More ID/OOD evaluation details and representative examples are provided in Appendix C**.

**Training without/with CoT.** For atomic facts ($S = S_{\text{ID}} \cup S_{\text{OOD}}$), training and evaluation are performed by having the model predict the final tail entity ($e_1, r_1 \xrightarrow{\text{predict}} \hat{e}_2, \forall (e_1, r_1, e_2) \in S$, $\hat{e}_2$ is the prediction of $e_2$, input $\xrightarrow{\text{predict}}$ output). As for two-hop facts, we consider whether to use CoT annotations during training ($(e_1, r_1, r_2, e_3) \in S_{\text{ID}_{\text{train}}}^{(2)}$). (1) Training without CoT: $e_1, r_1, r_2 \xrightarrow{\text{predict}} \hat{e}_3$, only predict the final tail entity $e_3$. (2) Training with CoT: $e_1, r_1, r_2 \xrightarrow{\text{predict}} \hat{e}_2$ and $e_1, r_1, r_2, \hat{e}_2 \xrightarrow{\text{predict}} \hat{e}_3$, predict both the bridge entity $e_2$ and the final tail entity $e_3$.

**Definition 3** (Training without/with CoT). *Let $\hat{e}$ denotes the output generated by the model $\mathcal{M}$, which is different from the ground truth. Taking two-hop facts $(e_1, r_1, r_2, e_3)$ as an example, the test loss is defined as $\mathbb{E}_{S_{\text{test}}^{(2)}}[\mathcal{L}(e_3, \mathcal{M}(e_1, r_1, r_2))]$, where $\mathcal{L}$ denotes the loss function. For training, (i) Training without CoT, the loss is $\mathbb{E}_{S_{\text{train}}^{(2)}}[\mathcal{L}(e_3, \mathcal{M}(e_1, r_1, r_2))]$; (ii) Training with CoT, the loss is $\mathbb{E}_{S_{\text{train}}^{(2)}}[\mathcal{L}(e_3, \mathcal{M}(e_1, r_1, r_2, \hat{e}_2)) + \mathcal{L}(e_2, \mathcal{M}(e_1, r_1, r_2))]$. In our analysis, we use the next-token prediction via autoregression [80, 52, 4] rather than next-token prediction via teacher-forcing [4].*

## 3.2 CoT Training Boosts Both ID and OOD Generalization

Although chain-of-thought (CoT) reasoning can improve task performance [44, 109, 101, 134, 56], it is absent during large-scale (pre-)training, when core model capabilities are developed [54, 138]. Prior work has extensively examined whether transformer-based language models can perform implicit composition, consistently reporting negative results [74, 117, 140]. Specifically, **for CoT prompting**, a persistent "compositionality gap" exists [74], where models often know all basic facts but fail to compose them—regardless of the model scale. **Regarding training without CoT**, Wang et al. [102] further demonstrate that transformers can learn to perform implicit reasoning under ID generalization, but this ability does not transfer to OOD settings. It naturally raises the question: **how does using explicit reasoning steps during training impact generalization ability?**

◇ **Controllable setup.** In order to control the data distribution, we utilize the data-generation process introduced in Section 3.1. Specifically, $S$ is constructed with $|\mathcal{E}| = 2000$ entities and $|\mathcal{R}| = 200$ relations (Appendix D.1). One-hop facts correspond to the $(e_1, r_1, e_2)$ triplets in $S$. These triplets are partitioned into two disjoint sets, that is, $S_{\text{ID}}$ (95%) and $S_{\text{OOD}}$ (5%), which are used to deduce the ID and OOD two-hop facts ($S_{\text{ID}}^{(2)}, S_{\text{OOD}}^{(2)}$). The candidate set of $\lambda$ is $\{0.001, 0.9, 1.8, 3.6, 7.2, 12.6\}$. Besides, the model employed is a standard decoder-only transformer as in GPT-2 [76], with a configuration of 8 layers, 768 hidden dimensions, and 12 attention heads, and the tokenization is done by having a unique token for each entity/relation for convenience[4].

---

[4]Details regarding the model, tokenization and optimization we used are provided in Appendix D.2.

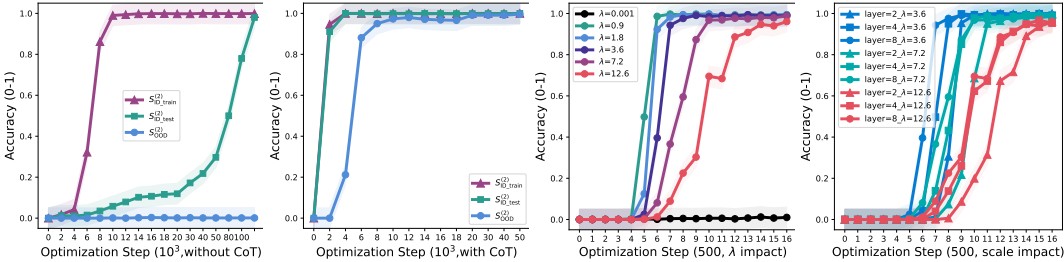

Figure 1: The model generalization ability under controllable data settings. **Left Part:** the accuracy comparison on two-hop reasoning between training without CoT (Left) and training with CoT (Center Left), $\lambda = 7.2$. CoT training significantly accelerates convergence and improves generalization from ID to OOD reasoning. **Right Part:** the impact of two-hop/one-hop ratio $\lambda$ (Center Right) and model scale (Right) on OOD generalization. Ratio $\lambda$ correlates with OOD generalization speed, while larger models converge more quickly without altering reasoning behavior.

$\diamond$ **Results.** Figure 1 (Left Part) illustrates model accuracy on the training and testing two-hop reasoning tasks during optimization with $\lambda = 7.2$. (1) **Training without CoT** (Left). We mirror and observe the same phenomenon (grokking [72]) as Wang et al. [102], namely that the model eventually generalizes to ID examples ($S_{\mathrm{ID_{test}}}^{(2)}$) only after extensive overtraining, while showing no OOD generalization ($S_{\mathrm{OOD}}^{(2)}$) even after millions of steps, suggesting delayed and non-systematic generalization likely based on *memorized patterns* (relation compositions). (2) **Training with CoT** (Center Left). Compared to training without CoT, incorporating CoT annotations significantly accelerates convergence and improves generalization. The model reaches near-perfect accuracy on $S_{\mathrm{ID_{test}}}^{(2)}$ within $\sim 4,000$ steps, and also shows substantial gains on $S_{\mathrm{OOD}}^{(2)}$. This indicates that explicit reasoning guidance during training shifts the model from nonsystematic (only ID) to more systematic (ID & OOD) generalization by enabling it to *learn underlying reasoning patterns* (relation compositions).

Ablation studies are also conducted to examine how the two-hop/one-hop ratio ($\lambda$), model scale influence CoT training performance, with a focus on accuracy over the OOD test set. (i) An appropriate $\lambda$ accelerates OOD generalization. Figure 1 (Center Right) shows the OOD test accuracy across different $\lambda$, which is strongly correlated with generalization speed. *A counter-intuitive finding is that a smaller ratio can expedite OOD generalization under CoT training[5], though overly small values may impair the model's ability to learn relevant reasoning patterns.* Indeed, the model may undergo a process of memorization before gradually learning the underlying patterns. In other words, a higher ratio increases the complexity of the training set, requiring a longer memorization phase, but the model eventually learns the relevant reasoning patterns. (ii) In Figure 1 (Right), we run the experiments with model layers $\in \{2, 4, 8\}$ and $\lambda \in \{3.6, 7.2, 12.6\}$, with other settings as in Figure 1 (Center Left). Scaling up the model does not qualitatively affect generalization, though larger models converge in fewer optimization steps, consistent with previous studies [54, 94, 102]. Additionally, we shift our focus to multi-hop scenarios in Appendix D.4, examining whether a model trained solely on two-hop facts during the CoT training phase can generalize to multi-hop facts.

### 3.3 MECHANISMS OF COT TRAINING: INTERNALIZING STEP-WISE REASONING

Up to this point, we have demonstrated that incorporating explicit CoT training in controlled experiments significantly enhances the model's reasoning generalization ability—shifting it from ID generalization alone to encompassing both ID and OOD generalization. Key factors such as data distribution (e.g., ratio $\lambda$ and pattern) play a crucial role in shaping the model's capacity for systematic generalization. **However, the internal mechanisms underlying these improvements remain unclear, which we aim to explore in this part**.

**Logit Lens & Causal Tracing.** Logit lens is widely used for inspecting hidden states of LLMs [12, 22, 40, 79, 117, 102]. The key idea behind the logit lens is to interpret intermediate hidden states by projecting them into the output vocabulary space using the model's output embedding ma-

---

[5]A key feature of reinforcement fine-tuning (RFT, used in OpenAI O1 models [67]) is its ability to train models for domain-specific tasks with minimal data, similar to our findings.

trix. We adopt the recent practice [117, 102] where the activation first goes through the transformer's final normalization layer before being multiplied by the output embedding. For causal tracing, the transformer can be interpreted as a causal graph [70] that propagates information from the input to the output via a network of intermediate states, enabling various causal analyses of its internal computations [98, 62, 104, 26, 18, 102]. Specifically, during the normal run, we intervene on the state of interest by replacing its activation with the activation from the perturbed run[6]. We then proceed with the remaining computations and measure whether the target state (the top-1 token through the logit lens) is altered. For simplicity, we denote a hidden state as $E[\text{layer index, token position}]$.

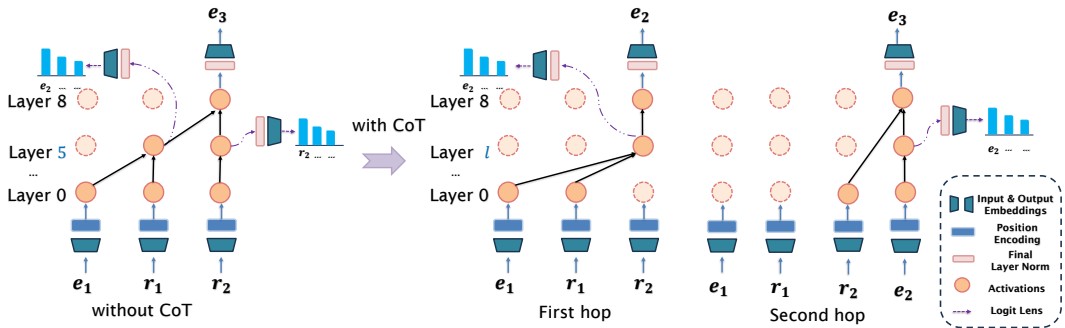

Figure 2: The compositional circuit (layer:8) for two-hop facts. We analyze individual states and assess the strength of connections between hidden states. **Left:** training without CoT, the circuit emerges only during ID generalization, with the intermediate result $e_2$ being resolved at layer index = 5. **Right:** training with CoT, the model achieves ID/OOD generalization via a two-stage circuit (sparse sequential dependencies between input tokens). It is noted that intermediate result $e_2$ is resolved at layer index = $l$, where $l = 3$ for ID and $l = 5$ for OOD.

We investigate the inner workings of the model during generalization using two widely adopted interpretability methods: logit lens [65] and causal tracing [70]. Specifically, we examine hidden-state representations by projecting intermediate-layer activations into the vocabulary space through the model's output embedding matrix. Besides, we assess the causal contribution of intermediate states by systematically replacing layer-wise representations, enabling us to identify the circuits responsible for generalization. Our analysis is conducted within the experimental setup described in Section 3.2. Here, we use $(e_1, r_1, e_2) \oplus (e_2, r_2, e_3) \implies (e_1, r_1, r_2, e_3), \forall e_1, e_2, e_3 \in \mathcal{E}, \forall r_1, r_2 \in \mathcal{R}$ to represent two-hop facts, in order to more clearly compare training with CoT and without CoT.

**Briefly, the circuit reflects the composition process**: CoT-trained models learn to systematically compose (both ID and OOD generalization) simpler skills ($P(C|X)$ & $P(Y|X, C)$), whereas those trained without CoT merely match patterns seen in the training data (only ID generalization).

⋄ **The internal mechanisms of training with CoT: two-stage compositional circuit.** We perform a set of logit lens and causal tracing experiments after training with CoT. Figure 2 (Right) illustrates the discovered compositional circuit, which represents the causal computational pathways after the 8-layer model achieves two-hop ID/OOD generalization. Specifically, we identify a highly interpretable causal graph consisting of states in layers 0, $l$, and 8, where weak nodes and connections have been pruned. The circuit exhibits two stages, consistent with the explicit reasoning steps in models during training. ❶ In the **first-hop stage**, layer $l$ splits the circuit into lower and upper parts: the lower parts retrieve the first-hop fact from the input $e_1, r_1, r_2$, store the bridge entity ($e_2$) in state $E[l, r_2]$; the upper parts pass the information of $e_2$ to output state $E[8, r_2]$ through the residual connections. Since the data distribution is controllable, $l$ can be precisely located (3 for $S_{\text{ID}_{\text{test}}}^{(2)}$, 5 for $S_{\text{OOD}}^{(2)}$). ❷ In the **second-hop stage**, the auto-regressive model uses the $e_2$ generated in the first-hop stage. This stage omits the $e_1, r_1$ and processes the second-hop from the input $e_1, r_1, r_2, e_2$ to store the tail $e_3$ to the output state $E[8, e_2]$. CoT training internalizes reasoning steps, with the number of reasoning circuit stages matching the number of explicit reasoning steps during training.

---

[6]If perturbing a node does not alter the target state (top-1 token through the logit lens), we prune the node. Please see method details in Appendix D.5.

◇ **The limits of training without CoT for OOD.** For training without CoT, we do the same circuit analysis and reproduce the results of Wang et al. [102]: the circuit emerges only during ID generalization. Layer 5 splits the circuit into lower and upper layers: the lower layers process the first-hop fact $(e_1, r_1, e_2)$ from the input $(e_1, r_1)$, storing the bridge entity $e_2$ in $E[5, r_1]$; the upper layers (5-8) gradually form the second-hop reasoning during extensive overtraining. Only when the intermediate result $e_2$ is resolved can the final result $e_1$ be correctly derived [6], traditional transformer cannot process subtasks in parallel. The difficulty in OOD generalization arises from semantic misalignment across layers, due to inconsistent token representations learned by the transformer [100].

◇ **How training with CoT mitigates transformer limits in OOD scenarios.** (1) As shown above, training with CoT allows the intermediate result $e_2$ to be extracted from a lower layer (index 3) for ID examples, whereas training without CoT requires a higher layer (index 5). *From an intuitive perspective, a smaller layer index implies that more layers remain available for processing the second hop, which could lead to better performance.* We also demonstrate that a two-layer transformer is sufficient to learn two-hop compositional circuits from CoT training in Appendix D.3. (2) CoT training explicitly processes each task through the model, allowing the model to independently learn each reasoning step. In brief, ID generalization is achievable in both CoT and non-CoT formats after sufficient training, but OOD generalization is challenging without CoT. *CoT enforces the decomposition of subtasks, effectively bringing OOD data closer to the ID, thereby helping the model learn a systematic generalization circuit.* This aligns the theoretical analysis in Section 2.

### 3.4 ROBUST SYSTEMATIC GENERALIZATION THROUGH CoT TRAINING

In practice, not all CoT training data is manually annotated, which means erroneous reasoning steps may be present. We aim to understand the source of CoT training's robustness under such imperfect conditions. In this section, we extend our analysis to settings where erroneous reasoning steps exist in the training data.

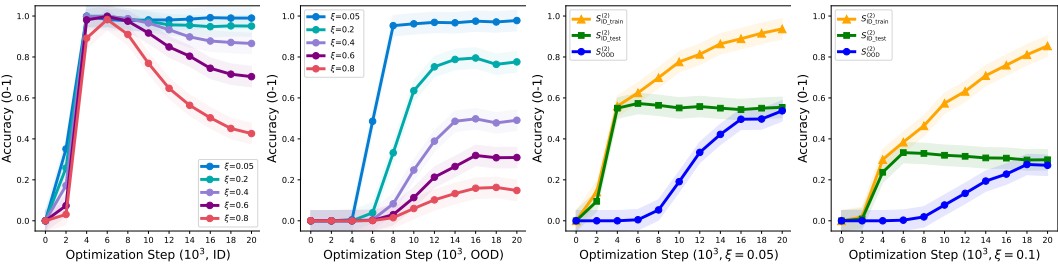

Figure 3: **Left Part:** the impact of *only* the second-hop noise on ID (Left) and OOD (Center Left) generalization. Noise has a significant impact on the final performance on both ID and OOD test data. However, the generalization trends for ID and OOD differ under noisy conditions. **Right Part:** the model's accuracy on training and testing two-hop reasoning facts at different noise ratios (*both* hops are noisy). It compares the results for $\xi$ values of 0.05 (Center Right), 0.1 (Right).

◇ **Method.** We aim to analyze the robustness of systematic generalization achieved through CoT training when noise is present in the training data. We use the same datasets as in Section 3.2, along with a two-layer model described in Appendix D.3. We introduce noise to the ID test examples $S_{\text{ID}_{\text{train}}}^{(2)}$ by randomly selecting a valid entity (the gold training target is $e_1, r_1, r_2, e_2, e_3$): (1) Only the second hop is noisy, $e_1, r_1, r_2, e_2, e_3^{\text{noise}}$; (2) Both hops are noisy, $e_1, r_1, r_2, e_2^{\text{noise}}, e_3^{\text{noise}}$. Note that the noise ratio is denoted as $\xi$, we explore the impact of different $\xi$. Next, we analyze different noise ratio ($\xi$) candidate sets for the two situations: $0.05, 0.2, 0.4, 0.6, 0.8$ for cases where only the second hop is noisy, and $0.05, 0.1$ for cases where both hops are noisy. More results are in Appendix D.6.

◇ **CoT training still enables systematic generalization even with data containing noisy steps** (In comparison to Figure 1 (Left)). (1) Figure 3 (Left Part) clearly demonstrates the impact of only the second-hop noise on ID and OOD generalization. Overall, under CoT training conditions, the model is still able to achieve systematic generalization from noisy training data, but its generalization ability decreases as the noise ratio increases. More specifically, as training progresses, OOD generalization initially remains constant and then increases, while ID generalization first increases

and then decreases. The decrease in ID generalization corresponds to the increase in OOD generalization. However, the final performance of both ID and OOD generalization decreases as the ratio increases. In particular, when the noise ratio ($\xi < 0.2$) is small, the model remains largely unaffected, highlighting the robustness of CoT training. Additionally, using the method from Section 3.3, we examine the circuits. Since noise is only introduced in the second hop, the first-hop circuit is well-learned, while the second-hop circuit is more significantly impacted by noise. (2) Figure 3 (Right Part) shows a comparison of the results for both hop noise $\xi$ values of $0.05, 0.1$. Adding noise to both hops has a much stronger suppressive effect on the model's generalization compared to adding noise only to the second hop. This indicates that the harm caused by CoT training data with completely incorrect reasoning steps is enormous. (3) To sum up, CoT training enables systematic generalization even with noisy training data, as long as the noise remains within a certain range[7]. Specifically, when the noise ratio is low, noisy data can still facilitate the model's learning of generalization circuits. The training bottleneck lies in collecting or synthesizing complex long CoT solutions, with some errors being acceptable. Additionally, we analyze the relationship between noisy samples and those with prediction errors, with further details on the effect of first-hop noise provided in Appendix D.7. Validations on real-world datasets (e.g., math word problems & compositional reasoning) are shown in Appendix E.

## 4 MORE DISCUSSION

**Theoretical Contribution**. According to the traditional statistical learning viewpoint, performance can be defined by the sum of optimization error and generalization error. For generalization (Section 2), the expected generalization error = population risk - empirical risk. More specifically, on the one hand, Theorems 1 and 2 show that CoT training helps reduce the expected generalization error. On the other hand, we can evaluate the empirical risk by assessing the model's performance on the validation set (Section 3.2). If the generalization error is known, it is possible to estimate the population risk. Therefore, the most challenging aspect is addressing the generalization error. For optimization, from an intuitive perspective, CoT-trained models learn to systematically compose simpler skills ($P(C|X), P(Y|X, C)$), whereas those trained without CoT merely match patterns ($P(Y|X)$) seen in the training data. That is, the optimization object for CoT training is $||P(C_{target}|X) - P(C_{predict}|X)||^2 + ||P(Y_{target}|X, C_{target}) - P(Y_{predict}|X, C_{predict})||^2$, which may be finer-grained (if the snowballing errors [4] are within an acceptable range) than optimization object for non-CoT training $||P(Y_{target}|X) - P(Y_{predict}|X)||^2$.

**Structural Discussion**. Our structural analysis (in Section 3.3) leads a discussion on the **key limitation of transformer** models in OOD generalization. For the circuit with two-hop facts $(e_1, r_1, e_2) \oplus (e_2, r_2, e_3) \rightarrow (e_1, r_1, r_2, e_3)$. (i) Training with CoT: the bridge entity ($e_2$) can be extracted from the middle layer hidden states $E(\text{layer index}, r_2)$, for ID layer index $= 3$, and for OOD, layer index $= 5$ (OOD is more challenging than ID). (ii) Training without CoT: The generalization circuit fails to emerge in OOD settings, aligning with the performance drop in Figure 1 (left). In ID settings, layer index $\geq 5$, higher than that in Training with CoT. **Intuitively, smaller layer index imply that more layers remain available for processing the second hop, potentially leading to better performance**. Only when the intermediate result $e_2$ is resolved can the final result $e_3$ be correctly derived, traditional transformer cannot process subtasks in parallel, which is the limitations of traditional transformer architectures.

The generalization circuit we discovered essentially corresponds to the reuse of the model's weights (effective depth increases [17]). Regarding the OOD generalization capability for compositional tasks, what we would truly like to see is whether the model can accomplish multi-step reasoning directly within a single computational process. To investigate this, we evaluate models trained with CoT—using twice the number of layers with weight reuse—under a modified input format: "$e_1, r_1, r_2, [\text{mask}]$". This setup is designed to emulate multiple forward passes typically required in next-token prediction. By examining the intermediate hidden states at the [mask] position (from the last layer), we observe that the model remains capable of correctly predicting $e_3$.

---

[7]Guo et al. [25] collect about 600k long CoT training samples. Errors during the process are acceptable, as it is impossible for humans to manually write 600k high-quality solutions. Moreover, through reinforcement learning, OpenAI O1 [68] learns to hone its chain of thought and refine the strategies it uses. It learns to recognize and correct its mistakes.

## 5 MORE RELATED WORK

**Chain-of-Thought Reasoning and Training.** Research has demonstrated that incorporating an intermediate reasoning process in language before producing the final output significantly enhances performance, especially for transformers [97] with advanced and robust generation capabilities. This includes prompting LLMs [109, 59, 139, 41, 81, 136, 93], or training LLMs to generate reasoning chains, either through supervised fine-tuning [132, 129] or reinforcement learning [105, 30, 85, 128, 126]. *However, the specific advantages of CoT training remain an open question, which is the focus of our study.* There are also theoretical analyses that have shown the usefulness of CoT from the perspective of model capabilities (e.g., expressivity) [17, 63, 53, 73, 127]. For instance, by employing CoT, the effective depth of the transformer increases as the generated outputs are fed back into the input [17]. *These analyses, along with the practice effectiveness of CoT, motivate our exploration of the core mechanisms underlying explicit CoT training. Our findings suggest that a two-layer transformer may be sufficient for learning compositional circuits through CoT training, which may explain the origin of CoT's expressivity during the training phase.*

**Latent Reasoning in Language Models.** The investigation of latent multi-hop abilities of LLMs could also have significant implications for areas such as generalization [45, 66] and model editing [137, 10]. To enhance the latent reasoning capabilities of LLMs, it is crucial to first understand the internal mechanisms by which LLMs effectively handle two-hop facts using latent multi-hop reasoning, which is also our focus. While a precise latent reasoning pathway has been found in controlled experiments [87, 64, 11, 7, 51, 77, 125, 102], it has not been thoroughly investigated in large pre-trained models. To fill this gap, Yang et al. [117] construct a dataset of two-hop reasoning problems and discovered that it is possible to recover the intermediate variable from the hidden representations. Biran et al. [6] identify a sequential latent reasoning pathway in LLMs, where the first-hop query is initially resolved into the bridge entity, which is then used to answer the second hop, they further propose to intervene the latent reasoning by "back-patching" the hidden representation. *Nevertheless, these studies focus only on CoT prompting and do not consider CoT training.* Recently, it has also been found that one can "internalize" the CoT reasoning into latent reasoning in transformers with knowledge distillation [13] or a special training curriculum that gradually shortens CoT [14]. Loop transformers [24, 8, 15] have been proposed to solve algorithmic tasks. *However, these works focus more on innovations in training methods and algorithms, without fully analyzing the underlying mechanisms of CoT training. This is precisely what we aim to uncover.*

## 6 CONCLUSION AND LIMITATION

**Conclusion.** In this work, we present a theoretical and structural framework to understand the generalization of CoT training. Our analysis reveals that the key mechanism behind CoT training lies in its ability to foster compositional generalization: models learn to systematically combine simpler, previously acquired skills to tackle novel and more complex problems. Theoretically, we demonstrate that the generalization error bound naturally decomposes into ID and OOD components. By breaking down complex tasks into simpler subtasks, CoT training narrows the gap between the training distribution and both ID and OOD test distributions, thereby enabling robust generalization performance. Structurally, we identify that CoT-trained models internalize reasoning into a staged compositional circuit—an architectural reflection of how CoT encourages models to learn "how to think", rather than merely "what to think". The quality of CoT data should be measured not only by the correctness of answers, but more critically, by the clarity and decomposition of key reasoning steps. Our findings have important implications for the design of CoT training strategies.

**Limitation.** However, further studies are required to (i) explore the potential of LLM reasoning in an unrestricted latent space, specifically through approaches such as training large language models to reason in a continuous latent space [29], (ii) extend our analysis to reinforcement fine-tuning settings [89, 82, 133, 21, 83] and (iii) while the notion of "simpler learned skills" is straightforward in synthetic tasks (e.g., atomic facts), it becomes substantially more implicit and difficult to delineate in real-world scenarios. Yuan et al. [131] propose that one practical approach is to first use a small seed set of SFT data to instill fundamental concepts within a target domain. Intuitively, the basic capabilities acquired by the model before CoT fine-tuning can be viewed as such "simpler learned skills". Those will deepen our understanding of the CoT training capabilities.

ACKNOWLEDGMENTS

This research was supported by National Key Research and Development Program of China (NO. 2024YFE0203200), Beijing Natural Science Foundation (Z250001), National Natural Science Foundation of China (No.62476277), CCF-ALIMAMA TECH Kangaroo Fund (No.CCF-ALIMAMA OF 2024008), and Huawei-Renmin University joint program on Information Retrieval. We also acknowledge the support provided by the fund for building worldclass universities (disciplines) of Renmin University of China and by the funds from Beijing Key Laboratory of Big Data Management and Analysis Methods, Gaoling School of Artificial Intelligence, Renmin University of China, from Engineering Research Center of Next-Generation Intelligent Search and Recommendation, Ministry of Education, from Intelligent Social Governance Interdisciplinary Platform, Major Innovation & Planning Interdisciplinary Platform for the "DoubleFirst Class" Initiative, Renmin University of China, from Public Policy and Decision-making Research Lab of Renmin University of China, and from Public Computing Cloud, Renmin University of China.

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

## LLM USAGE

Regarding the use of LLMs, they were employed solely for language polishing purposes and played no role in research ideation, literature retrieval, or any other academically substantive activities.

## A MORE RELATED WORK

**Chain-of-Thought Reasoning.** Research has demonstrated that incorporating an intermediate reasoning process in language before producing the final output significantly enhances performance, especially for transformers [97] with advanced and robust generation capabilities. This includes prompting LLMs [109, 59, 139, 41, 81, 136, 93], or training LLMs to generate reasoning chains, either with supervised fine-tuning [132, 129, 123] or reinforcement learning [105, 30, 85, 128, 126, 124]. There are also theoretical analyses have shown the usefulness of CoT from the perspective of model expressivity and generalization [17, 63, 53, 73, 122, 120, 118, 119, 127, 37, 36]. By employing CoT, the effective depth of the transformer increases as the generated outputs are fed back into the input [17]. This aligns with our findings, suggesting that a two-layer transformer is sufficient for learning compositional circuits through CoT training. While CoT has proven effective for certain tasks, its autoregressive generation nature makes it difficult to replicate human reasoning on more complex problems [46, 27], which often require planning and search [113, 121, 28, 47, 20, 88, 49]. These analyses, along with the practice effectiveness, motivate our exploration of the core mechanisms underlying CoT training.

**Latent Reasoning in Language Models.** The investigation of latent multi-hop abilities of LLMs could also have significant implications for areas such as generalization [45, 66] and model editing [137, 10]. To enhance the latent reasoning capabilities of LLMs, it is crucial to first understand the internal mechanisms by which LLMs effectively handle two-hop facts using latent multi-hop reasoning, which is also our focus. While a precise latent reasoning pathway has been found in controlled experiments [87, 64, 11, 7, 51, 77, 125, 102], it has not been thoroughly investigated in large pre-trained models. To fill this gap, Yang et al. [117] construct a dataset of two-hop reasoning problems and discovered that it is possible to recover the intermediate variable from the hidden representations. Biran et al. [6] identify a sequential latent reasoning pathway in LLMs, where the first hop query is initially resolved into the bridge entity which is then used to answer the second hop, they further propose to intervene the latent reasoning by "back-patching" the hidden representation. Shalev et al. [84] discover parallel latent reasoning paths in LLMs. Ghandeharioun et al. [23] introduce a framework called "Patchscopes" to explain LLMs internal representations in natural language. Recently, it has also been found that one can "internalize" the CoT reasoning into latent reasoning in the transformer with knowledge distillation [13] or a special training curriculum which gradually shortens CoT [14]. Loop transformers [24, 8, 15] have been proposed to solve algorithmic tasks. These have some similarities to the two-stage compositional circuit we present.

# B OMITTED PROOFS AND DEFINITIONS

## B.1 LEMMA 1 AND PROOF

**Lemma 1** (KL Divergence Decomposition). *Under the conditions of Assumption 1, the KL divergence between test and training conditional distributions satisfies:*

$$D_{KL}(P_{test}(Y|X)||P_{train}(Y|X)) = - H(\alpha) + (1 - \alpha)D_{KL}(P_{test}^{ID}(Y|X)||P_{train}(Y|X))$$
$$+ \alpha D_{KL}(P_{test}^{OOD}(Y|X)||P_{train}(Y|X)),$$

*where $D_{KL}(P_{test}(Y|X)||P_{train}(Y|X)) = \mathbb{E}_{y|x \sim P_{test}} \left[ \log \frac{P_{test}(Y|X)}{P_{train}(Y|X)} \right]$. $H(\alpha)$ is the entropy of $\alpha$ quantifying ID/OOD uncertainty, the other terms capture ID/OOD deviation from the training distribution.*

*Proof.* Original definition of KL divergence:

$$D_{KL}(P_{test}(Y|X)||P_{train}(Y|X)) = \mathbb{E}_{y|x \sim P_{test}} \left[ \log \frac{P_{test}(Y|X)}{P_{train}(Y|X)} \right]$$

Since $P_{test}$ is a mixture distribution, the expectation can be decomposed into two parts:

$$= (1 - \alpha) \, \mathbb{E}_{y|x \sim P_{test}^{ID}} \left[ \log \frac{P_{test}(Y|X)}{P_{train}(Y|X)} \right] + \alpha \, \mathbb{E}_{y|x \sim P_{test}^{OOD}} \left[ \log \frac{P_{test}(Y|X)}{P_{train}(Y|X)} \right]$$

Due to the disjoint support assumption:

$$= (1 - \alpha) \, \mathbb{E}_{y|x \sim P_{test}^{ID}} \left[ \log \frac{(1 - \alpha)P_{test}^{ID}(Y|X)}{P_{train}(Y|X)} \right] + \alpha \, \mathbb{E}_{y|x \sim P_{test}^{OOD}} \left[ \log \frac{\alpha P_{test}^{OOD}(Y|X)}{P_{train}(Y|X)} \right]$$

By isolating the constant term and the KL divergence:

$$= \alpha \log \alpha + (1 - \alpha) \log(1 - \alpha) + (1 - \alpha)D_{KL}(P_{test}^{ID}(Y|X)||P_{train}(Y|X))$$
$$+ \alpha D_{KL}(P_{test}^{OOD}(Y|X)||P_{train}(Y|X))$$
$$= - H(\alpha) + (1 - \alpha)D_{KL}(P_{test}^{ID}(Y|X)||P_{train}(Y|X))$$
$$+ \alpha D_{KL}(P_{test}^{OOD}(Y|X)||P_{train}(Y|X)),$$

where $H(\alpha) = - \sum_{i=1}^{K} \alpha_i \log \alpha_i, \sum_{i=1}^{K} \alpha_i = 1, \forall \alpha_i \geq 0$. This completes the proof. $\square$

## B.2 EXPECTED GENERALIZATION ERROR

Recently, information-theoretic generalization bounds [114, 78, 86, 108, 107] have been introduced to analyze the expected generalization error of learning algorithms. A key benefit of these bounds is that they depend not only on the data distribution but also on the specific algorithm, making them an ideal tool for studying the generalization behavior of models trained using particular algorithms.

**Definition 4** (Expected Generalization Error). *We let $\mathcal{Z} = \mathcal{X} \times \mathcal{Y}$ be the instance space and $\mu$ be an unknown distribution on $\mathcal{Z}$, specifying random variable $Z$. Here, $\mathcal{X}$ denotes the feature space and $\mathcal{Y}$ is the label space. Suppose one observes a training set $S_N \triangleq (Z_1, ..., Z_N) \in \mathcal{Z}^N$, with $N$ i.i.d. training examples drawn from $\mu$. In the information-theoretic analysis framework, we let $\mathcal{W}$ be the space of hypotheses related to the model, and a stochastic learning algorithm $\mathcal{A}$ which takes the training examples $S_N$ as its input and outputs a hypothesis $W \in \mathcal{W}$ according to some conditional distribution $Q_{W|S_N}$. Given a loss function $\ell : \mathcal{W} \times \mathcal{Z} \to \mathbb{R}^+$, where $\ell(w, Z)$ measures the "unfitness" or "error" of any $Z \in \mathcal{Z}$ with respect to a hypothesis $w \in \mathcal{W}$. We take $\ell$ as a continuous function and assume that $\ell$ is differentiable almost everywhere with respect to $w$. The goal of learning is to find a hypothesis $w$ that minimizes the population risk, and for any $w \in \mathcal{W}$, the population risk is defined as $L_\mu(w) \triangleq \mathbb{E}_{Z \sim \mu}[\ell(w, Z)]$. However, since only can partially observe $\mu$ via the sample $S_N$, we instead turn to use the empirical risk, defined as $L_{S_N}(w) \triangleq \frac{1}{N} \sum_{i=1}^{N} \ell(w, Z_i)$. Then the expected generalization error of $\mathcal{A}$ is defined as*

$$\widetilde{error} \triangleq \mathbb{E}_{W, S_N}[L_\mu(W) - L_{S_N}(W)],$$

*where the expectation is taken over $(S_N, W) \sim \mu^N \otimes Q_{W|S_N}$.*

**Remark 3.** *At this point, revisiting Definition 1 and Assumption 1, $S_N$ corresponds to the training data distribution $P_{train}(Y|X)$, while $\mu$ corresponds to the test data distribution $P_{test}(Y|X)$. Accordingly, the equation can be reformulated as:*

$$\widetilde{error} \triangleq \mathbb{E}[h(S^{test}, W')] - \mathbb{E}[h(S_N^{train}, W)],$$

*where $h(s, w) = \frac{1}{N} \sum_{i=1}^{N} l(w, Z_i)$.*

*That is, generalization error=population risk-empirical risk.*

### B.3 PROOF OF THEOREM 1

Information-theoretic generalization bounds commonly center on input-output mutual information, with the Donsker-Varadhan representation of KL divergence (Polyanskiy and Wu [71, Theorem 3.5]) serving as a key analytical foundation.

**Lemma 2** (Donsker and Varadhan's variational formula). *Let $P_1, P_2$ be probability measures on $\mathcal{Z}$, for any bounded measurable function $f : \mathcal{Z} \to \mathbb{R}$, we have $D_{KL}(P_1||P_2) = \sup_f \mathbb{E}_{Z \sim P_1}[f(Z)] - \log \mathbb{E}_{Z \sim P_2}[\exp f(Z)]$.*

**Lemma 3** (Wang [107, Lemma 2.4.6]). *Let $P_1$ and $P_2$ be probability measures on $\mathcal{Z}$. Let $Z' \sim P_1$ and $Z \sim P_2$. If $g(Z)$ is $R$-subgaussian, then,*

$$|\mathbb{E}_{Z' \sim P_1}[g(Z')] - \mathbb{E}_{Z \sim P_2}[g(Z)]| \leq \sqrt{2R^2 D_{KL}(P_1||P_2)}.$$

*Proof.* Let $f = t \cdot g$ for any $t \in \mathbb{R}$, by Lemma 2, we have

$$D_{\text{KL}}(P_1||P_2) \geq \sup_t \mathbb{E}_{Z' \sim P_1}[tg(Z')] - \log \mathbb{E}_{Z \sim P_2}[\exp t \cdot g(Z)]$$

$$= \sup_t \mathbb{E}_{Z' \sim P_1}[tg(Z')] - \log \mathbb{E}_{Z \sim P_2}[\exp t(g(Z) - \mathbb{E}_{Z \sim P_2}[g(Z)] + \mathbb{E}_{Z \sim P_2}[g(Z)])]$$

$$= \sup_t \mathbb{E}_{Z' \sim P_1}[tg(Z')] - \mathbb{E}_{Z \sim P_2}[tg(Z)] - \log \mathbb{E}_{Z \sim P_2}[\exp t(g(Z) - \mathbb{E}_{Z \sim P_2}[g(Z)])]$$

$$\geq \sup_t t(\mathbb{E}_{Z' \sim P_1}[g(Z')] - \mathbb{E}_{Z \sim P_2}[g(Z)]) - t^2 R^2/2,$$

where the last inequality is by the subgaussianity of $g(Z)$.

Then consider the case of $t > 0$ and $t < 0$ ($t = 0$ is trivial), by AM–GM inequality (i.e. the arithmetic mean is greater than or equal to the geometric mean), the following is straightforward,

$$|\mathbb{E}_{Z' \sim P_1}[g(Z')] - \mathbb{E}_{Z \sim P_2}[g(Z)]| \leq \sqrt{2R^2 D_{\text{KL}}(P_1||P_2)}.$$

This can also be proven by computing the extremum through differentiation with respect to $t$. $\square$

Below, we provide the proof of Theorem 1.

*Proof.* Apply Lemma 3 to the generalization error (Definition 4), we let $g(s, w) = \frac{1}{N} \sum_{i=1}^{N} l(w, Z_i)$. Thus, if $l(w, Z)$ is $R$-subgaussian for all $w \in \mathcal{W}$, then $g(S_N^{\text{train}}, W)$ is $R/\sqrt{N}$-subgaussian due to the i.i.d. assumption on $Z_i$'s. This implies that $g(S^{\text{test}}, W')$ is also $R/\sqrt{N}$-subgaussian, where $S^{\text{test}}, W'$ are independent copies of $S_N^{\text{train}}, W$, respectively. So we have:

$$\widetilde{error} \leq \sqrt{\frac{2R^2}{N} D_{\text{KL}}(P_{\text{test}}||P_{\text{train}})}, \tag{1}$$

According to Lemma 1, for $-H(\alpha) \leq 0$:

$$D_{\text{KL}}(P_{\text{test}}||P_{\text{train}}) = -H(\alpha) + (1 - \alpha)D_{\text{KL}}(P_{\text{test}}^{\text{ID}}||P_{\text{train}}) + \alpha D_{\text{KL}}(P_{\text{test}}^{\text{OOD}}||P_{\text{train}})$$

$$\leq (1 - \alpha)D_{\text{KL}}(P_{\text{test}}^{\text{ID}}||P_{\text{train}}) + \alpha D_{\text{KL}}(P_{\text{test}}^{\text{OOD}}||P_{\text{train}}).$$

Putting everything together, we arrive at:

$$\widetilde{error} \leq \sqrt{\frac{2R^2}{N} \left[(1 - \alpha)D_{\text{KL}}(P_{\text{test}}^{\text{ID}}||P_{\text{train}}) + \alpha D_{\text{KL}}(P_{\text{test}}^{\text{OOD}}||P_{\text{train}})\right]}.$$

Under the conditions of Lemma 1 and Lemma 3, we complete the proof. $\square$

## B.4 Extensions of Theorem 1

Recall Theorem 1, the expected generalization error is bounded by:

$$\widetilde{\text{error}} \leq \sqrt{\frac{2R^2}{N} \left[ (1-\alpha) D_{\text{KL}}(P^{\text{ID}}_{\text{test}}(Y|X) || P_{\text{train}}(Y|X)) + \alpha D_{\text{KL}}(P^{\text{OOD}}_{\text{test}}(Y|X) || P_{\text{train}}(Y|X)) \right]},$$

where $N$ is the training data size, $Z = (X, Y)$ and $\mathcal{W}$ is the space of hypotheses related to the model. $\alpha$ denotes the mixing coefficient of the OOD data in test distribution and $D_{\text{KL}}(\cdot)$ represents the KL divergence between ID/OOD test and training distributions.

For ID, since $S^{(2)}_{\text{ID}_{\text{test}}}$ consists of unseen instances within known compositions $(r_i, r_j)$ from $S^{(2)}_{\text{ID}_{\text{train}}}$, the reasoning patterns in ID test examples exactly match those in the training set. Therefore, with sufficient training—whether with or without CoT—the ID generalization error approaches zero (i.e., $D_{\text{KL}}(P^{\text{ID}}_{\text{test}} || P_{\text{train}}) = 0$). Consequently, we obtain the following corollary:

**Corollary 1.** *Let the conditions specified in Theorem 1 hold, and assume sufficient training such that $D_{KL}(P^{ID}_{test} || P_{train}) \to 0$. Then the expected generalization error is dominated by the OOD component:*

$$\widetilde{error} \leq \sqrt{\frac{2R^2\alpha}{N} D_{KL}(P^{OOD}_{test}(Y|X) || P_{train}(Y|X))}.$$

*This implies that when ID patterns are perfectly learned, the error scales with $\sqrt{\alpha D^{OOD}_{KL}}$, revealing that OOD generalization fundamentally limits performance even under optimal ID training.*

For OOD, considering the data generation process, we have: $P(Y|X) = \sum_C P(Y|X,C)P(C|X)$, where $C$ denotes the CoT reasoning steps. (a) In training without CoT, due to the model does not explicitly learn $P(C|X)$ (i.e., $C$ never appears during training), the $P(C|X)$ becomes a uniform prior, making $P(Y|X,C)$ reduce to $P(Y|X)$. Moreover, since OOD test examples $S^{(2)}_{\text{OOD}_{\text{test}}}$ involve unseen compositional patterns, training without CoT struggles to generalize to OOD settings (the reasoning circuit only emerges during ID generalization). **That is, under training without CoT, there has been no improvement in the upper bound of OOD generalization.**

(b) In contrast, under training with CoT, both $P(C|X)$ and $P(Y|X,C)$ are explicitly modeled, which correspond precisely to the two stages of the compositional circuit. Since the sub-tasks have been observed during training, when the step-wise decomposition in $S^{(2)}_{\text{ID}_{\text{train}}}$ and $S^{(2)}_{\text{OOD}}$ aligns, OOD generalization can also achieve near-perfect performance. The next result provides an explicit form.

**Corollary 2** (Thereom 2, OOD Generalization Error for Training with CoT)**.** *Let the conditions specified in Theorem 1 hold, and assume sufficient training such that $D_{KL}(P^{ID}_{test} || P_{train}) \to 0$. Define $P(Y|X) = \sum_C P(Y|X,C)P(C|X)$, then the OOD generalization error is bounded by:*

$$\begin{aligned}
\widetilde{error}^2 \leq \frac{2R^2\alpha}{N} [ & D_{KL}(P^{OOD}_{test}(C|X) || P_{train}(C|X)) \\
& + \mathbb{E}_{C \sim P^{OOD}_{test}(C|X)}[D_{KL}(P^{OOD}_{test}(Y|X,C) || P_{train}(Y|X,C))]],
\end{aligned}$$

*where $C_i$ denotes the CoT reasoning steps, see Appendix B.5 for a proof.*

**Remark 4.** *When the intermediate reasoning outcomes (e.g., data patterns and reasoning steps) from explicit CoT training are highly aligned with or closely match the intermediate reasoning required for final inference during testing, the model's generalization ability is significantly enhanced. Specifically, the model explicitly learns both $P(C|X)$ and $P(Y|X,C)$, such that $D_{KL}(P^{OOD}_{test}(C|X) || P_{train}(C|X)) \to 0$ and $D_{KL}(P^{OOD}_{test}(Y|X,C) || P_{train}(Y|X,C)) \to 0$. Therefore, OOD generalization can also approach optimal performance under CoT-based training.*

**Remark 5** (Length Generalization Discussion)**.** *Length generalization continues to be an indispensable research avenue in CoT investigations [140, 126, 112, 1, 106], we also experimentally study in Appendix D.4. From a theoretical perspective (Corollary 2), models achieve optimal generalization capability when the number of reasoning steps during training ($m_1$) matches that during testing ($m_2$), as this allows the model to independently learn to optimize the $D_{KL}(\cdot)$ for each individual step. The model can still generalize well when $m_1 > m_2$ and $m_2$ is a prefix structure [106] of $m_1$ (i.e., the subtasks in the shorter chain have all been encountered in the longer chain). On the other hand, when $m_1 < m_2$ and $m_2$ contains unseen subtasks, generalization becomes more challenging.*

### B.5 PROOF OF THEOREM 2

We are given two conditional distributions defined as mixtures:

$$P_1(y \mid x) = \sum_c P_{11}(y \mid c, x) P_{12}(c \mid x), \quad P_2(y \mid x) = \sum_c P_{21}(y \mid c, x) P_{22}(c \mid x).$$

We aim to derive the KL divergence between $P_1(y \mid x)$ and $P_2(y \mid x)$:

$$D_{\mathrm{KL}}(P_1 \| P_2) = \sum_y P_1(y \mid x) \log \frac{P_1(y \mid x)}{P_2(y \mid x)}.$$

Substituting the expressions for $P_1$ and $P_2$:

$$D_{\mathrm{KL}}(P_1 \| P_2) = \sum_y \left[ \sum_c P_{11}(y \mid c, x) P_{12}(c \mid x) \right] \log \frac{\sum_{c'} P_{11}(y \mid c', x) P_{12}(c' \mid x)}{\sum_{c'} P_{21}(y \mid c', x) P_{22}(c' \mid x)}$$

$$= \sum_y \sum_c P_{12}(c \mid x) P_{11}(y \mid c, x) \log \frac{\sum_{c'} P_{11}(y \mid c', x) P_{12}(c' \mid x)}{\sum_{c'} P_{21}(y \mid c', x) P_{22}(c' \mid x)}.$$

This is the exact form of the KL divergence but is difficult to simplify further due to the logarithm over sums. We now derive a useful upper bound using the log-sum inequality ($f(x) = x \log x$ is a convex function):

$$\sum_i a_i \log \frac{a_i}{b_i} \geq \left( \sum_i a_i \right) \log \frac{\sum_i a_i}{\sum_i b_i}.$$

Let

$$a_i = P_{11}(y \mid i, x) P_{12}(i \mid x), \quad b_i = P_{21}(y \mid i, x) P_{22}(i \mid x).$$

Then the inequality implies:

$$P_1(y \mid x) \log \frac{P_1(y \mid x)}{P_2(y \mid x)} \leq \sum_c P_{12}(c \mid x) P_{11}(y \mid c, x) \log \frac{P_{11}(y \mid c, x) P_{12}(c \mid x)}{P_{21}(y \mid c, x) P_{22}(c \mid x)}.$$

Summing both sides over $y$:

$$D_{\mathrm{KL}}(P_1 \| P_2) \leq \sum_{y,c} P_{12}(c \mid x) P_{11}(y \mid c, x) \left[ \log \frac{P_{11}(y \mid c, x)}{P_{21}(y \mid c, x)} + \log \frac{P_{12}(c \mid x)}{P_{22}(c \mid x)} \right]$$

$$= \sum_c P_{12}(c \mid x) \sum_y P_{11}(y \mid c, x) \log \frac{P_{11}(y \mid c, x)}{P_{21}(y \mid c, x)}$$

$$+ \sum_c P_{12}(c \mid x) \log \frac{P_{12}(c \mid x)}{P_{22}(c \mid x)}$$

$$= \mathbb{E}_{c \sim P_{12}(\cdot \mid x)} \left[ D_{\mathrm{KL}}(P_{11}(\cdot \mid c, x) \| P_{21}(\cdot \mid c, x)) \right] + D_{\mathrm{KL}}(P_{12}(\cdot \mid x) \| P_{22}(\cdot \mid x)).$$

Thus, we obtain the following upper bound:

$$\boxed{D_{\mathrm{KL}}(P_1 \| P_2) \leq D_{\mathrm{KL}}(P_{12}(\cdot \mid x) \| P_{22}(\cdot \mid x)) + \mathbb{E}_{c \sim P_{12}(\cdot \mid x)} \left[ D_{\mathrm{KL}}(P_{11}(\cdot \mid c, x) \| P_{21}(\cdot \mid c, x)) \right]}$$

Apply it to Corollary 1, we complete the proof of Theorem 2 (Corollary 2).

## B.6 THEOREM 3 AND PROOF

Given that the reasoning pattern remained unchanged during the noise injection process in Sections 3.4 and E, we adopt the following assumption regarding the training data distribution:

**Assumption 2** (Noise Injection to Training Distribution). *Assume that output noise randomly replaces the original outputs with probability $\xi$. Let the original output distribution be denoted as $P_{train}(Y|X)$, then, the output distribution after noise injection is given by:*

$$P_{new}(Y|X) = (1 - \xi)P_{train}(Y|X) + \xi P_{noise}(Y|X),$$

*where $P_{noise}(Y|X)$ represents the distribution over outputs after random replacement.*

**Lemma 4** (Convexity of the KL divergence). *For a fixed probability distribution $P$, the KL divergence $D_{KL}(P||Q)$ is a convex function with respect to $Q$. That is, for any two distributions $Q1, Q2$, and any $\beta \in [0, 1]$, the following inequality holds:*

$$D_{KL}(P||(1 - \beta)Q_1 + \beta Q_2) \leq (1 - \beta)D_{KL}(P||Q_1) + \beta D_{KL}(P||Q_2).$$

*Proof.* The KL divergence is defined as:

$$D_{KL}(P \| Q) = \sum_x P(x) \ln \frac{P(x)}{Q(x)}.$$

Now consider $Q = (1 - \beta)Q_1 + \beta Q_2$. Substituting this into the definition of KL divergence gives:

$$D_{KL}(P \| (1 - \beta)Q_1 + \beta Q_2) = \sum_x P(x) \ln \frac{P(x)}{(1 - \beta)Q_1(x) + \beta Q_2(x)}.$$

Using the convexity of the logarithmic function $\ln(1/t)$ (since $\ln(t)$ is concave, and $\ln(1/t) = -\ln(t)$ is therefore convex), we have:

$$\ln \frac{P(x)}{(1 - \beta)Q_1(x) + \beta Q_2(x)} \leq (1 - \beta) \ln \frac{P(x)}{Q_1(x)} + \beta \ln \frac{P(x)}{Q_2(x)}.$$

Multiplying both sides by $P(x)$ (noting that $P(x) \geq 0$) and summing over $x$, we obtain:

$$\sum_x P(x) \ln \frac{P(x)}{(1 - \beta)Q_1(x) + \beta Q_2(x)} \leq (1 - \beta) \sum_x P(x) \ln \frac{P(x)}{Q_1(x)} + \beta \sum_x P(x) \ln \frac{P(x)}{Q_2(x)}.$$

That is, $D_{KL}(P||(1 - \beta)Q_1 + \beta Q_2) \leq (1 - \beta)D_{KL}(P||Q_1) + \beta D_{KL}(P||Q_2)$. $\square$

**Theorem 3** (Generalization Bounds with Noise Ratio). *Under the conditions specified in Assumption 2 and Definition 4, we assume that the loss $\ell(w, Z)$ is R-subGaussian for any $w \in \mathcal{W} \in \mathbb{R}^d$, then the expected generalization error is bounded by:*

$$\widetilde{error} \leq \sqrt{\frac{2R^2}{N} [(1 - \xi)D_{KL}(P_{test}||P_{train}) + \xi D_{KL}(P_{test}||P_{noise})]},$$

*where $N$ is the training data size, $Z = (X, Y)$ and $\mathcal{W}$ is the space of hypotheses related to the model. $\xi$ denotes the noise ratio in training distribution and $D_{KL}(\cdot)$ represents the KL divergence between two distributions. $P_{test} = (1 - \alpha)P_{test}^{ID} + \alpha P_{test}^{OOD}$.*

*Proof.* Replace $P_{train}(Y|X)$ in Eq. (1) with $P_{new}(Y|X)$, we have:

$$\widetilde{error} \leq \sqrt{\frac{2R^2}{N} D_{KL}(P_{test}||P_{new})}.$$

To apply Assumption 2 and Lemma 4, we let $\beta = \xi, P = P_{test}, Q = P_{new}$, that is, $Q_1 = P_{train}, Q_2 = P_{noise}$. Therefore, we will get:

$$\widetilde{error} \leq \sqrt{\frac{2R^2}{N} [(1 - \xi)D_{KL}(P_{test}||P_{train}) + \xi D_{KL}(P_{test}||P_{noise})]}.$$

We complete the proof. $\square$

## C  EVALUATION AND REPRESENTATIVE EXAMPLES

**ID/OOD Evaluation.**  To better evaluate the generalization capacity of the model, we assess its performance on both ID and OOD data. (1) ID generalization aims to determine whether the model has correctly learned the latent patterns by evaluating its ability to complete previously unseen two-hop facts $S^{(2)}_{\text{ID}_{\text{test}}}$. (2) OOD generalization aims to assess the systematicity [44] acquired by the model, specifically its ability to apply learned patterns to knowledge irrespective of its distribution. This is done by testing the model on facts $S^{(2)}_{\text{OOD}}$. If the model performs well on ID data, it may have memorized or learned patterns present in the training data $T$. **However, strong performance on OOD data indicates that the model has learned the latent patterns, as $T$ contains only atomic facts $S_{\text{OOD}}$, not $S^{(2)}_{\text{OOD}}$.** Representative examples highlighting the distributional differences between ID and OOD data are presented below.

The key distinction between ID and OOD data lies in whether their distributions are the same. We provide a more prominent example with distribution differences here. The training set is composed of $S_{\textbf{ID}}$, $S_{\textbf{OOD}}$ and $S^{(2)}_{\textbf{ID}_{\textbf{train}}}$.

$S_{\textbf{ID}}$:
$(\text{Paris}, \text{CapitalOf}, \text{France}), (\text{France}, \text{LocatedIn}, \text{Europe}),$
$(\text{Berlin}, \text{CapitalOf}, \text{Germany}), (\text{Germany}, \text{LocatedIn}, \text{Europe}).$

$S_{\textbf{OOD}}$:
$(\text{Lima}, \text{CapitalOf}, \text{Peru}), (\text{Peru}, \text{HasNaturalFeature}, \text{Andes Mountains}).$

$S^{(2)}_{\textbf{ID}_{\textbf{train}}}$, a uniformly random subset of the inferred facts derived from $S_{\text{ID}}$:
$(\text{Paris}, \text{CapitalOfCountryLocatedIn}, \text{Europe}).$

$S^{(2)}_{\textbf{ID}_{\textbf{test}}}$, previously unseen inferred facts derived from $S_{\text{ID}}$ (ID generalization):
$(\text{Berlin}, \text{CapitalOfCountryLocatedIn}, \text{Europe}).$

$S^{(2)}_{\textbf{OOD}}$, previously unseen inferred facts derived from $S_{\text{OOD}}$ (OOD generalization):
$(\text{Lima}, \text{CapitalOfCountryWithNaturalFeature}, \text{Andes Mountains}).$

**Key Changes.**  This OOD data setup requires the model to tackle greater knowledge distribution differences and reasoning challenges during evaluation. (1) Change in Relation Types: New relation types, such as "HasNaturalFeature," are introduced in the OOD data. (2) Complexity of Reasoning Paths: Reasoning paths involving natural features are added, requiring the model not only to reason but also to generalize to new types of knowledge.

## D  MORE EXPERIENCE RESULTS

### D.1  DATA GENERATION PROCESS

Specifically, for one-hop facts, a random knowledge graph $\mathcal{G}$ is constructed with $|\mathcal{E}|$ entities and $|\mathcal{R}| = 200$ relations. Each entity, acting as the head ($h$), is linked to 20 unique relations, with each relation connecting to another randomly selected entity serving as the tail ($t$). One-hop facts correspond to the $(h, r, t)$ triplets in $\mathcal{G}$. $S \subset \mathcal{G} \subset \mathcal{S}$. These triplets are partitioned into two disjoint sets: $S_{\text{ID}}$ (95%) and $S_{\text{OOD}}$ (5%), which are used to deduce the ID/OOD two-hop facts ($\forall h, b, t \in \mathcal{E}, \forall r_1, r_2 \in \mathcal{R}$): $(h, r_1, b) \oplus (b, r_2, t) \Longrightarrow (h, r_1, r_2, t)$.

### D.2  TRAINING DETAILS OF CONTROLLABLE DATA

**Model.**  The model we employ is a standard decoder-only transformer as in GPT-2 [76]. We use 8-layer for Figure 1 (Left, Center Left, Center Right) and Figure 2, while 2,4,8-layer for Figure 1 (Right) and 2-layer for Figures 3 and 4. The other hyperparameters are consistent with those described in Section 3.1.

**Tokenization.**  In our main content (Section 3.2), we assign a unique token to each relation/entity by default. This is because Wang et al. [102] find that different tokenizations affect the results in

rather expected ways, and do not influence the reasoning generalization findings. We also validate our findings in the real-world entity-tokenization in Section E.

**Optimization.** Optimization is done by AdamW [57] with learning rate $10^{-4}$, batch size 512, weight decay 0.1 and 2000 warm-up steps. Notably, models are trained for a large number of epochs/steps beyond the point where training performance saturates. All runs were done with PyTorch [69] and Huggingface Transformers [111] on NVIDIA GeForce RTX 3090.

### D.3 TWO-LAYER MODEL CIRCUIT

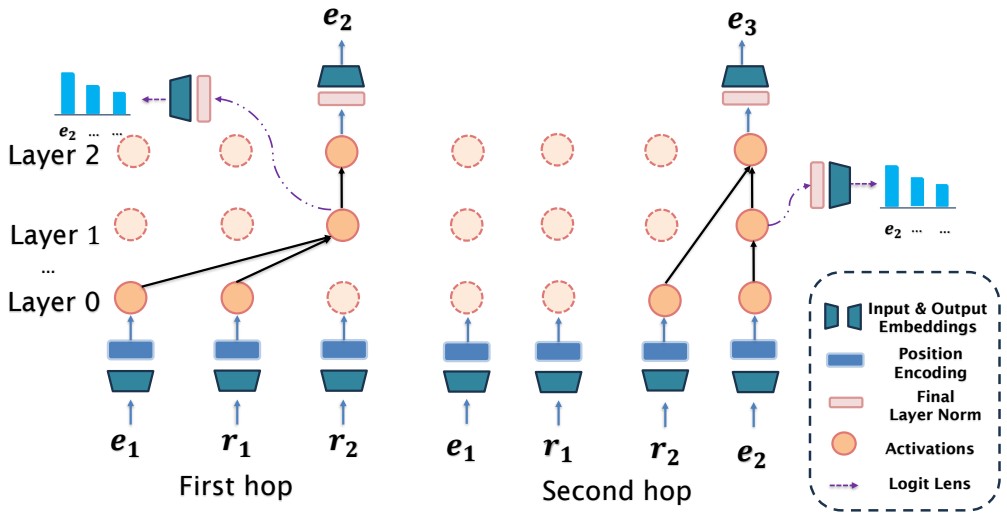

Figure 4: The two-stage compositional circuit (layer:2) for two-hop facts. We use logit lens to interpret individual states, and use causal tracing to measure the strength of connections between states. **It is evident that a two-layer model can learn compositional circuits from CoT training**. This aligns with Cabannes et al. [8], who describe how a certain distribution of weights within the first two attention layers of a transformer, referred to as an "iteration head," enables a transformer to solve iterative tasks with CoT reasoning with relative ease. We identify CoT reasoning circuits in a transformer with only 2 layers. The output of first stage is autoregressively used for the second.

For CoT prompting, the effective depth of the transformer increases because the generated outputs are looped back to the input [17].

### D.4 TWO-HOP TO MUTI-HOP

In Sections 3.2 to 3.3, we primarily focus on two-hop facts. In this subsection, we shift our focus to multi-hop scenarios: Can a model that has only encountered two-hop facts during the CoT training phase generalize to three-hop facts[8]?

**Experiment Setup.** The rule of (three-hop) composition is $(e_1, r_1, e_2) \oplus (e_2, r_2, e_3) \oplus (e_3, r_3, e_4) \implies (e_1, r_1, r_2, r_3, e_4), \forall e_1, e_2, e_3, e_4 \in \mathcal{E}, \forall r_1, r_2, r_3 \in \mathcal{R}$. In CoT training, we only use one-hop/two-hop facts $(e_1, r_1 \xrightarrow{\text{predict}} e_2; e_1, r_1, r_2 \xrightarrow{\text{predict}} e_2, e_3)$, and we test whether the model can generalize to reasoning in three-hop facts $(e_1, r_1, r_2, r_3 \xrightarrow{\text{predict}} e_2, e_3, e_4)$. Other settings are same in Section 3.1.

---

[8]It can naturally extend to multi-hop scenarios once thoroughly analyzed.

**Results.** A model that has only encountered two-hop data during CoT training cannot directly generalize to three-hop scenarios. *Intuitively, if we consider $e_1, r_1, r_2 \rightarrow e_2, e_3$ and $e_2, r_2, r_3 \rightarrow e_3, e_4$ as new atomic facts and $e_1, r_1, r_2, r_3 \rightarrow e_2, e_3, e_4$ as their compositional combination, models trained exclusively on atomic facts exhibit limited reasoning capability for compositional facts.* However, when we artificially split a three-hop fact into two two-hop facts for testing, the model is also able to generalize effectively. In other words, we test $e_1, r_1, r_2 \xrightarrow{\text{predict}} e_2, e_3$ and $e_2, r_2, r_3 \xrightarrow{\text{predict}} e_3, e_4$ separately, and when both are correct, we consider $e_1, r_1, r_2, r_3 \xrightarrow{\text{predict}} e_2, e_3, e_4$ to be correct. These align with [8]: CoT relates to reproducing reasoning patterns that appear in the training set.

**Discussion.** For cite references, Zhu et al. [140] emphasize that the model fails to directly deduce $A_i \rightarrow C_i$ when the two intermediate steps $A_i \rightarrow B_i$, $B_i \rightarrow C_i$ are trained separately, this corresponds to that the absence of three-hop instances in the training data significantly limits its generalization capability to three-hop queries during evaluation. Moreover, when we include three-hop facts in the training set (|three-hop : one-hop| = 3.6), the model achieves over 0.9 accuracy on both ID and OOD test sets. Building upon this, we further conduct circuit analysis on the model trained with three-hop facts, revealing a three-stage mechanism: the first stage activates $e_1, r_1$, the second stage activates $r_2, e_2$, and the third stage activates $r_3, e_3$. Besides, we agree with [140] that "the reversal curse is a consequence of the (effective) model weights asymmetry", autoregressive models struggle with parallel generation of subtasks (see Section 3.3 for details). The generalization circuit we discovered essentially corresponds to the **reuse of the model's weights**. We also note the emergence of "Loop-transformer Reasoning" [130] works. Our results further support the validity of this research direction.

> **Insight:** When the intermediate reasoning results (data patterns) from explicit CoT training are highly aligned with or closely match the intermediate reasoning required for final inference during testing, the model's generalization ability is significantly enhanced. This could explain why the researchers behind DeepSeek-R1 [25] construct and collect a small amount of *long CoT* data to fine-tune the model in the *Cold Start* phase.

## D.5 CAUSALITY ANALYSIS DETAILS

**The methodological descriptions here serve to clarify the approach we used, not to attribute these methods to our own work.** We mirror the recent practice [104, 102], where the causal tracing process consists of three steps (for every stage in Section 3.3):

(1) In the normal run, we capture the model's hidden state activations for a regular input $(e_1, r_1, r_2)/(e_1, r_1, r_2, e_2)$. Notably, as the model maintains perfect training accuracy throughout the CoT training process, the final prediction invariably aligns with the ground truth[9] —— bridge entity $e_2$ for the first-hop stage and tail entity $e_3$ for the second-hop stage.

(2) During the perturbed run[10], the model is given a slightly modified input that alters its prediction, and the hidden state activations are recorded once again.

Specifically, for the hidden state of interest, we modify the input token at the corresponding position by substituting it with a random alternative of the same type (e.g., $r_1 \rightarrow r_1^{'}$), which results in a different target prediction (e.g., $e_2 \rightarrow e_2^{'}, e_3 \rightarrow e_3^{'}$).

(3) Intervention. During the normal run, we intervene the activation of the state of interest by substituting it with its corresponding activation from the perturbed run. After completing the remaining computations, we check whether the target state (determined as the top-1 token via the logit lens) has changed. If perturbing a node does not alter the target state (top-1 token through the logit lens), we prune the node.

---

[9]For simplicity, when we refer to a state as a token, we are indicating the top-ranked token of that state as determined by the logit lens.

[10]For the perturbation, studies have explored adding noise to the input [62] and replacing key tokens with semantically close ones [98, 18]. We adopt token replacement which avoids unnecessary distribution shifts [135].

## D.6 CoT Training with Noise

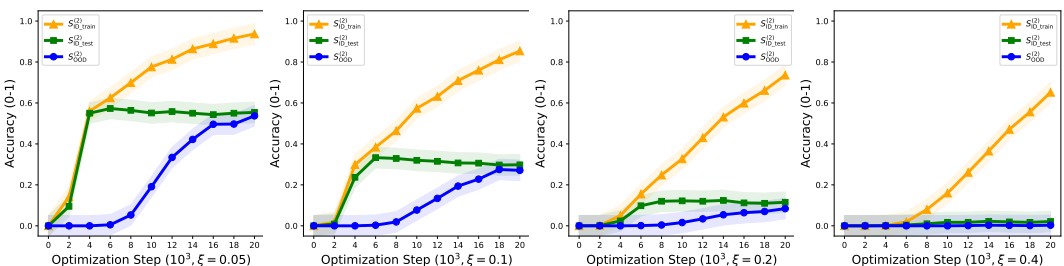

Figure 5: The model's accuracy on training and testing two-hop reasoning facts at different noise ratios (both hops are noisy). This figure compares the results for $\xi$ values of $0.05, 0.1, 0.2,$ and $0.4$.

Table 1: The model's accuracy (100%) on all reasoning facts at different noise ratios (only the second hop is noisy). This table compares the results for $\xi$ values of 0.05, 0.2, 0.4, 0.6 and 0.8 with 20,000 optimization steps in Figure 3 (Left Part).

| $\xi$ | $S_{\text{ID}}$ | $S_{\text{OOD}}$ | $S_{\text{ID}_{\text{TRAIN}}}^{(2)}$ | $S_{\text{ID}_{\text{TEST}}}^{(2)}$ | $S_{\text{OOD}}^{(2)}$ |
|---|---|---|---|---|---|
| 0.05 | 99.9 | 100.0 | 99.3 | 99.0 | 97.8 |
| 0.2 | 100.0 | 99.9 | 98.9 | 95.1 | 77.6 |
| 0.4 | 100.0 | 99.9 | 98.0 | 86.6 | 49.1 |
| 0.6 | 100.0 | 99.5 | 96.8 | 70.4 | 30.9 |
| 0.8 | 100.0 | 99.2 | 97.3 | 42.6 | 14.8 |

Table 2: The model's accuracy (100%) on all reasoning facts at different noise ratios (both the first and the second hops are noisy). This table compares the results for $\xi$ values of 0.05, 0.1, 0.2 and 0.4 with 20,000 optimization steps in Figure 5.

| $\xi$ | $S_{\text{ID}}$ | $S_{\text{OOD}}$ | $S_{\text{ID}_{\text{TRAIN}}}^{(2)}$ | $S_{\text{ID}_{\text{TEST}}}^{(2)}$ | $S_{\text{OOD}}^{(2)}$ |
|---|---|---|---|---|---|
| 0.05 | 98.3 | 99.7 | 93.7 | 55.4 | 53.7 |
| 0.1 | 97.5 | 97.4 | 85.4 | 29.8 | 27.1 |
| 0.2 | 95.0 | 92.2 | 73.6 | 11.5 | 8.4 |
| 0.4 | 84.5 | 72.4 | 65.2 | 2.1 | 0.3 |

**Results.** We analyze different $\xi$ (noise ratio) candidate sets for the two situations: $\{0.05, 0.2, 0.4, 0.6, 0.8\}$ for only the second hop is noisy and $\{0.05, 0.1, 0.2, 0.4\}$ for both hops are noisy. The comparison results are as follows:

(1) Figure 3 and Table 1 clearly demonstrate the impact of only the second-hop noise on ID and OOD generalization ($S_{\text{ID}_{\text{test}}}^{(2)}$ and $S_{\text{OOD}}^{(2)}$). Overall, under CoT training conditions, the model is still able to achieve systematic generalization from noisy training data, but its generalization ability decreases as the noise ratio increases. More specifically, as training progresses, OOD generalization initially remains constant and then increases, while ID generalization first increases and then decreases. The decrease in ID generalization corresponds to the increase in OOD generalization. However, the final performance of both ID and OOD generalization decreases as the ratio increases. In particular, when the noise ratio ($\xi < 0.2$) is relatively small, the model is almost unaffected, demonstrating the robustness of CoT training. Furthermore, we also use the method in Section 3.3 to examine the circuits. Since we only add noise in the second hop, the circuit in the first-hop stage is learned relatively well, while the circuit in the second-hop stage is more heavily affected by noise.

(2) Figure 5 and Table 2 show a comparison of the results for both hop noise $\xi$ values of $0.05, 0.1, 0.2,$ and $0.4$. Adding noise to both hops has a much stronger suppressive effect on the model's generalization compared to adding noise only to the second hop. A noise ratio greater than 0.2 is enough to nearly eliminate both ID and OOD generalization ability.

(3) Table 1 and Table 2 demonstrate that adding noise to both hops exerts a significantly stronger suppressive effect on the model's generalization compared to introducing noise solely to the second hop. When the noise ratio is 0.2, the accuracy of $S_{\text{ID}_{\text{test}}}^{(2)}$ in Table 1 is 95.1%, and the accuracy of $S_{\text{OOD}}^{(2)}$ is 77.6%. However, in Table 2, the accuracies of $S_{\text{ID}_{\text{test}}}^{(2)}$ and $S_{\text{OOD}}^{(2)}$ are only 11.5% and 8.4%. This indicates that the harm caused by CoT training data with completely incorrect reasoning steps is enormous.

(4) To sum up, even with noisy training data, CoT training can still enable the model to achieve systematic generalization when the noise is within a certain range[11]. Especially when the noise ratio is small, these noisy data can still help the model learn generalization circuits. Moreover, we further analyze the connection between noisy samples and the samples with prediction errors from the model, and give the impact of only the first-hop is noisy in Appendix D.7.

> **Insight:** CoT training still enables systematic generalization with noisy data, highlighting that data quality outweighs the method itself. The training bottleneck lies in collecting or synthesizing complex *long CoT* solutions, with some errors being acceptable.

## D.7   NOISE DISCUSSION

In Section 3.4, we demonstrated that adding noise to both hops exerts a significantly stronger suppressive effect on the model's generalization compared to introducing noise solely to the second hop. Here, we further investigate the impact of scenarios where only the first hop is noisy. We introduce noise to $S_{\text{ID}_{\text{train}}}^{(2)}$ by randomly selecting a valid entity, where the gold training target is $e_1, r_1, r_2, e_2, e_3$. In this case, only the first hop is noisy, resulting in $e_1, r_1, r_2, e_2^{\text{noise}}, e_3$.

**Only the First-Hop is Noisy.** Table 3 highlights that as $\xi$ increases, performance generally declines across all datasets, with the most significant drops observed in $S_{\text{ID}_{\text{test}}}^{(2)}$ and $S_{\text{OOD}}^{(2)}$ at higher noise levels. For instance, when $\xi = 0.4$, the accuracy for $S_{\text{OOD}}^{(2)}$ falls to $0.3\%$, indicating a substantial degradation in model generalization under noisy conditions. This degradation indicates that errors in intermediate steps within the CoT training data have a greater impact. However, the model still retains a certain level of generalization capability under low noise conditions.

Table 3: The model's accuracy (100%) on all reasoning facts at different noise ratios (only the first hop is noisy). This table compares the results for $\xi$ values of 0, 0.05, 0.1, 0.2, and 0.4. Regarding the optimization (20,000 steps) and model settings, we remain consistent with Section 3.4. All values are averaged over 3 random seeds.

| $\xi$ | $S_{\text{ID}}$ | $S_{\text{OOD}}$ | $S_{\text{ID}_{\text{TRAIN}}}^{(2)}$ | $S_{\text{ID}_{\text{TEST}}}^{(2)}$ | $S_{\text{OOD}}^{(2)}$ |
|---|---|---|---|---|---|
| 0 | 100.0 | 99.9 | 99.9 | 100.0 | 99.4 |
| 0.05 | 98.4 | 99.4 | 94.3 | 53.3 | 51.3 |
| 0.1 | 97.9 | 98.2 | 86.3 | 29.7 | 29.5 |
| 0.2 | 97.0 | 94.5 | 74.9 | 10.6 | 7.5 |
| 0.4 | 89.6 | 77.0 | 68.7 | 1.8 | 0.3 |

**Noisy Samples Affect the Probability Distribution of the Output Vocabulary Space.** As shown in Table 3, adding noise to $S_{\text{ID}_{\text{train}}}^{(2)}$ can even impact the model's basic knowledge ($S_{\text{ID}}$ and $S_{\text{OOD}}$). We further analyze the relationship between noisy samples and the samples with prediction errors from the model. Namely, we use a sample $e_{1567}, r_{32}, r_{23} \xrightarrow{\text{gold label}} e_{1002}, e_{404}$ to validate the model after CoT training with $\xi = 0.05$. The model's erroneous output is $e_{1567}, r_{32}, r_{23} \xrightarrow{\text{wrong predict}} e_{1640}, e_{665}$, and we identify the noisy sample in the training set as: $e_{1567}, r_{32}, r_{132} \xrightarrow{\text{noisy label}} e_{1640}, e_{851}$.

---

[11]Guo et al. [25] collect about 600k long CoT training samples. Errors during the process are acceptable, as it is impossible for humans to manually write 600k high-quality solutions. Moreover, through reinforcement learning, OpenAI O1 [68] learns to hone its chain of thought and refine the strategies it uses. It learns to recognize and correct its mistakes.

We check the probability distribution of the output vocabulary space: (1) For one-hop fact $e_{1567}, r_{32}(\xrightarrow{\text{gold label}} e_{1002})$, the token with the highest output probability is $e_{1002}$, followed by $e_{1640}$ as the second highest. Adding excessive noise can bias the previously learned one-hop facts. (2) For two-hop fact $e_{1567}, r_{32}, e_{\text{any1}}, (\xrightarrow{\text{gold label}} e_{1002}, e_{\text{any2}})$, the token with the highest output probability is $e_{1640}$, followed by $e_{1002}$ as the second highest. The structure of such noisy data can indeed contribute to the learning of generalization performance. However, the addition of noise can negatively impact the reasoning of related facts.

### D.8 DETAILS OF REAL-WORLD DATA VERIFICATION

**Data Descriptions.** The test data is sourced from GSM8K [9] (containing 1,319 samples), while the training data is derived from MetaMathQA [129] (comprising 395K samples), which enhances data quality and improves model performance. Considering training cost and following Meng et al. [61], we use only the first 100k samples for 1 epoch of training.

**Training Formats.** The data has been converted into alpaca format [91] for ease of use. Unlike the non-CoT format, which simply outputs "The Answer is ...", the CoT format provides reasoning steps alongside the answer.

**Hyperparameter Configurations** We employ LoRA [34] with rank 16 and alpha 16, without dropout (0). The model is trained for 1 epoch with a maximum sequence length of 512 tokens. The training uses a batch size of 4 per device without gradient accumulation (steps=1). The optimization employs a learning rate of 2e-5 with no weight decay (0) and a warmup ratio of 0.03. The framework and hardware information we used is provided in Appendix D.2.

We randomly inject noise with a probability of noise ratio. Notably, as shown in Table 4 (Right), the model was trained for only 1,000 steps.

**Noise Types.** Only two types of noise are added: (1) Step skipping (replacing intermediate steps with "..."). (2) Incorrect step results (keeping the left-hand expression while modifying the right-hand result). Below we show an example comparing the results before and after noise injection.

```
sample_data = {
'output': 'At first, there were 10 snowflakes.\nEvery 5 minutes, an
    additional x snowflakes fell, so after t minutes, there were 10 + (t
    /5)*x snowflakes.\nWe are given that there were 58 snowflakes, so we
    can write: 10 + (t/5)*x = 58.\nSolving for t, we get: (t/5)*x = 48.\
    nDividing both sides by x, we get: t/5 = 48/x.\nMultiplying both
    sides by 5, we get: t = (5*48)/x.\nWe are given that t = 60 minutes,
    so we can write: (5*48)/x = 60.\nSolving for x, we get: x = (5*48)
    /60.\nSimplifying the right side, we get: x = 4.\nThe value of x is
    4.\n#### 4\nThe answer is: 4',
'instruction': 'There were 10 snowflakes at first...',
'type': 'GSM_FOBAR',
'input': ''
}
noise_data = {
'output': 'At first, there were 10 snowflakes.\n'
'Every 5 minutes, an additional x snowflakes fell, so after t minutes,
    there were 10 + (t/5)*x snowflakes.\n'
'We are given that there were 58 snowflakes, so we can write: 10 + (t/5)*
    x = 57.\n'  # Incorrect step results (58->57)
'Solving for t, we get: (t/5)*x = ...\n'  # Step skipping
'Dividing both sides by x, we get: t/5 = 49/x.\n'  # Incorrect step
    results (48->49)
'Multiplying both sides by 5, we get: t = (6*48)/x.\n'  # Incorrect step
    results (5->6)
'We are given that t = 60 minutes, so we can write: (5*48)/x = 59.\n'  #
    Incorrect step results (60->59)
'Solving for x, we get: x = (5*48)/61.\n'  # Incorrect step results
    (60->61)
'Simplifying the right side, we get: x = ...\n'  # Step skipping
```

```
'The value of x is 4.\n'
'#### 4\n'
'The answer is: 4',
'instruction': 'There were 10 snowflakes at first...',
'type': 'GSM_FOBAR',
'input': ''}
```

## D.9 LAYER-WISE GRADIENTS CHANGE

Figure 6: The nuclear norm of gradients across different layers when training with CoT. This aligns with the Figure 1 (Right), we plot the setting with layer $= 4, \lambda = 7.2$. The checkpoint=4 means the optimization steps=4,000, when the OOD test accuracy begins to grow. The ID generalization phase $(0-4)$ leads to larger gradients across layers compared to the OOD generalization phase $(5-8)$. This aligns with the results [49], indicating that slow thinking with CoT results in smaller gradients and enables the gradients to effectively distinguish between correct and irrelevant reasoning paths.

**Nuclear Norm:** The nuclear norm of gradient $(G)$ is defined as the $l_1$ norm of the singular values, which reflects the sparsity of the spectrum and serves as a convex surrogate of the matrix rank. Hence, it does not only quantify the gradient magnitude but also the concentration of the spectrum on its top few singular values, which is vital to understand the gradient patterns in each layer, i.e., $||G||_{\text{nuclear}} = \sum_{j=1}^{\min(m,n)} |\sigma_j|, \sigma_j(j = 1, ..., )$ are singular values of $G$.

# E    REALISTIC DATA VERIFICATION

At a high level, our study so far paves the way for a deeper understanding and enhancement of the transformer's generalization abilities through CoT training on controlled data distributions. However, real-world training data distributions are often more complex. Accordingly, we empirically validate the insights presented in Sections 3.2 and 3.4 on real-world datasets.

◇ **Experiment setup with real-world data.** Following[12] prior work [129, 61], we perform LoRA-based fine-tuning [34] (rank=16) of LLaMA3.1-8B [3] and Qwen2.5-3B [92] using MetaMathQA [129], then evaluate their mathematical reasoning performance on the GSM8K [9] test set. (1) For training without CoT, the model's target consists solely of the final answer, whereas training with CoT incorporates the reasoning steps. (2) To avoid alterations to the reasoning format, we only inject noise into the mathematical expression components of reasoning steps, randomly selecting between two noise types: a) step skipping only; b) incorrect step results. The details of data descriptions, training formats, noise types and hyperparameter configurations are provided in Appendix D.8.

Table 4: **Left:** GSM8K performance with/without CoT fine-tuning (bold: best results). **Right:** the impact of mathematical expression noise ratio. The Qwen2.5-3B is fine-tuned for only 1000 steps to reduce training overhead. Even with the noise ratio=1, the model maintains improved reasoning generalization through fine-tuning with CoT.

| Model | Method | GSM8K (100%) | Qwen2.5-3B | |
|---|---|---|---|---|
| | | | Noise Ratio | GSM8K (100%) |
| Llama3.1-8B | before fine-tune | 33.13 | 0 | 69.37 |
| | fine-tuning without CoT | 22.79 | | |
| | fine-tuning with CoT | **74.01** | 0.4 | 69.06 |
| Qwen2.5-3B | before fine-tune | 14.56 | 0.8 | 68.98 |
| | fine-tuning without CoT | 19.21 | 1.0 | 68.83 |
| | fine-tuning with CoT | **78.81** | | |

◇ **Verification Results.** (1) Table 4 (Left) shows that CoT fine-tuning significantly improves accuracy on GSM8K benchmark compared to answer-only fine-tuning and pre-trained baselines, even fine-tuning without CoT may degrade the original reasoning capabilities of the base model. For instance, LLaMA3.1-8B's performance dropped from 33.13 to 22.79 after answer-only fine-tuning. (2) In Table 4 (Right), since the original reasoning patterns in the training data are preserved and noise is injected solely into the mathematical expression components of the reasoning steps, the model achieves improved reasoning generalization through CoT fine-tuning. Notably, fine-tuning with CoT (noise ratio = 1) results in a significantly higher accuracy of 68.83%, compared to only 19.21% for fine-tuning without CoT. Together, these findings substantiate the claims made in Sections 3.2 and 3.4 with solid empirical evidence. Additional real-world validations are also provided in Appendix E.1.

---

[12]Considering training cost, this part uses only the first 100k samples for only one epoch of training like [61].

### E.1 MORE REALISTIC DATA RESULTS

**Experiment Setup.** We extend our experiments to Llama2-7B [95] to verify whether the insights (that the model can achieve systematic composition generalization through CoT training) hold true for real-world datasets. We use the dataset provided by [6], which contains 82,020 two-hop queries based on data from Wikidata [99]. To exclude the influence of the model's inherent knowledge, we filter the data and split the training and test set, the filtering process can be seen as follows (**the methods described clarify our approach and data, without implying originality**):

Referring to Biran et al. [6], we aim to filter out cases where no latent reasoning occurs. Given a two hop query $((e_1, r_1, e_2), (e_2, r_2, e_3)$ we test two prompts constructed to detect and filter out cases where the model performs reasoning shortcuts [115, 103, 38]. (1) The first prompt is $((, r_1, e_2), (e_2, r_2, e_3)$ (i.e., the query without $e_1$), aimed at filtering out cases where the model predicts generally popular entities. (2) The second prompt is $((e_1, , e_2), (e_2, r_2, e_3)$ (i.e., the query without $r_1$), aimed at filtering out cases where the model predicts correctly due to a high correlation between $e_1$ and $e_3$. We perform this filtering for the model (Llama2-7B(-chat)) using greedy decoding creating a subset of the dataset.

We are specifically interested in the model's reasoning performance, therefore, we further generate two dataset subsets: (1) The first subset is made up of 6,442 cases where the model correctly answers both the two-hop query and the first hop. (2) The second subset includes 2,320 cases where the model correctly answers both the first and second hop in isolation, but fails to answer the full two-hop query. Notice that we only use the first subset for CoT training and the second subset for generalization evaluation. When it is about to training and testing data: (1) The training set is made up of 6,442 cases where the model correctly answers both the two-hop query and the first hop. (2) The test set includes 2,320 cases where the model correctly answers both the first and second hop in isolation, but fails to answer the full two-hop query.

We force the model to only output the answers to both hops, for example, the model input is "the capital of the country of the base of OpenAI is" and the model target is "the United States. Washington, D.C." To be more specific, the first hop of the example is "the country of the base of OpenAI is the United States." and the second hop of the example is "the capital of the United States is Washington, D.C.". Due to resource limitations, we fine-tune the model for 100 epochs on the training set using LoRA [34] (rank=32) and test it on the test set. The framework and hardware information we used is provided in Appendix D.2.

In more detail, we examined the outputs of the model after fine-tuning. Excitingly, when the model receives the input "the capital of the country of citizenship of Jenna Jameson is" (in test set), the first token it outputs is "Washington.", which is also the first token of "Washington, D.C". This indicates that CoT training does indeed provide the model with some generalization ability.

