# OpenReview forum: "Compositional Generalization from Learned Skills via CoT Training: A Theoretical and Structural Analysis for Reasoning"
_ICLR.cc/2026/Conference — ICLR 2026 Poster_

### Official Review · Reviewer_ECGg · 2025-10-19

**Soundness:** 2
**Presentation:** 3
**Contribution:** 2
**Rating:** 4
**Confidence:** 5

**Summary:**

This paper proposes that Chain-of-Thought (CoT) training enhances out-of-distribution generalization by enabling models to systematically combine learned skills—a process termed compositional reasoning. This claim is buttressed by both theoretical generalization bounds and controlled experiments. Furthermore, a circuit analysis uncovers the internal, multi-stage computational structure that underlies this ability.

**Strengths:**

1. This paper provides both theoretical analysis and experimental results to support their claims and a mechanism analysis is also completed, which makes the paper comprehensive.
2. The use of a controlled synthetic task allows for a crisp demonstration of the core principle—compositional generalization.
3. The presentation of this paper is clear and easy to understand.

**Weaknesses:**

1. In my view, the results achieved by the CoT training paradigm proposed in the paper still essentially rely on the model performing single-step reasoning. The model merely learns to recognize the first two tokens in the pattern "e1, r1, r2" and complete the first step of reasoning, and then, given the input "e1, r1, r2, e2", it learns to recognize the last two tokens and complete the second step. What the model has actually learned are two distinct single-step reasoning tasks under different input formats, rather than demonstrating genuine compositional reasoning ability. Moreover, this two-step prediction approach is essentially no different from performing single-step reasoning on atomic data and then explicitly executing two separate single-step inferences. Regarding the OOD generalization capability for compositional tasks, what we would truly like to see is whether the model can accomplish multi-step reasoning directly within a single computational process.  Figure 1 (center right) may also be interpreted by this concern, that there is a trade-off between learning two-hop formations and the atomic reason.
2. There is a discernible disconnect between the theoretical framework and the experimental work. The theoretical results remain without a concrete experimental counterpart for their verification.
3. The robustness analysis feels somewhat detached from the preceding sections. Furthermore, the findings presented in Figure 3 lack a control group for comparison, and the observed trends are intuitively self-evident, which collectively diminishes the insightfulness of this investigation.
4. The related work section discusses latent reasoning, yet I fail to see which part of this paper actually relates to it. If the proposed CoT method is indeed performed in a latent space, the authors should explicitly clarify this point in method part. In that case, I would reconsider my first weakness and the overall contribution of this work.

**Questions:**

See weakness.

---

> ### Author Response · Authors · 2025-11-16
> **Thank you! (part 1)**
>
> Dear Reviewer ECGg,
>
> Thank you for your careful evaluation and thoughtful suggestions. We deeply value your time and effort in providing such constructive feedback. Below, we address your comments point by point. We also have revised the manuscript and marked the changes in **orange**.
>
> > **W1**: In my view, the results achieved by the CoT training paradigm proposed in the paper still essentially rely on the model performing single-step reasoning. The model merely learns to recognize the first two tokens in the pattern "e1, r1, r2" and complete the first step of reasoning, and then, given the input "e1, r1, r2, e2", it learns to recognize the last two tokens and complete the second step. What the model has actually learned are two **distinct single-step** reasoning tasks under different input formats, rather than demonstrating genuine compositional reasoning ability. Moreover, this two-step prediction approach is essentially no different from performing single-step reasoning on atomic data and then explicitly executing two separate single-step inferences. Regarding the OOD generalization capability for compositional tasks, what we would truly like to see is whether the model can accomplish multi-step reasoning directly within a single computational process. Figure 1 (center right) may also be interpreted by this concern, that there is a trade-off between learning two-hop formations and the atomic reason.
>
> > **A**: We sincerely appreciate your thorough comment. We are glad to have the opportunity to clarify.
> >
> > (1) **There may be some misunderstandings regarding the CoT training paradigm**. In our Section 3.1 "Training with CoT: $e_1, r_1, r_2 \xrightarrow{\text{predict}} \hat{e}_2 \text{ and } e_1, r_1, r_2, \hat{e}_2 \xrightarrow{\text{predict}} \hat{e}_3$, predict both the bridge entity $e_2$ and the final tail entity $e_3$". Also in Section 3.3 "In the second-hop stage, **the auto-regressive model uses the $\hat{e}_2$ generated in the first-hop stage."  We did not use distinct single-step prompts (such as "e1, r1, r2" and "e1, r1, r2, e2") as you mentioned, but strictly adhered to the next-token prediction paradigm [1,2]** (that is $x_t=M(x_1,...,x_{t-1})$). This **does not contradict** the demonstration of genuine compositional reasoning ability. (We have updated Lines 226-235.)
> >
> > (2) For the OOD generalization capability for compositional tasks.  **Owing to the next-token prediction paradigm, the claim of "accomplishing multi-step reasoning directly within a single computational process"** appears to be inappropriate. However, we will explore this further in our discussion on **A to W4**.
> >
> > (3) **We agree with your professional observation** "that there is a trade-off between learning two-hop formations and the atomic reason". As we shown in Section 3.2 for Figure 1, "Indeed, the model may undergo a process of memorization before gradually learning the underlying patterns. In other words, a higher ratio increases the complexity of the training set, requiring a longer memorization phase, but the model eventually learns the relevant reasoning patterns".
> >
> > [1] Towards Understanding the Universality of Transformers for Next-Token Prediction. ICLR 2025.
> >
> > [2] On the Generalization Ability of Next-Token-Prediction Pretraining. ICML 2025.
>
> > **W2**: There is a discernible disconnect between the theoretical framework and the experimental work. The theoretical results remain without a concrete experimental counterpart for their verification.
>
> > **A**: We sincerely thank the reviewer for raising this important point. We appreciate the opportunity to further summarize and clarify our work's contribution. (Section 5 in the revised version)
> >
> > (1) Theoretically, as defined in Section 2, **the expected generalization error  = population risk - empirical risk**. More specifically, on the one hand, Theorems 1/2 show that CoT training helps reduce the **expected generalization error**. On the other hand, we can evaluate the **empirical risk** by assessing the model's performance on the validation set (for example, Figure 1). If the generalization error is known, it is possible to estimate the **population risk**. Therefore, the key aspect is addressing the generalization error.
> >
> > (2) Empirically, as we stated in Section 3, our experiments serving as empirical support for the aforementioned theoretical claims.  Please let we point one by one: Figures 1 and 2 support Theorems 1 and 2 (ID&OOD discussion, compositional circuit), Figure 3 supports Theorem 3 (noise discussion).

---

> > ### Author Response · Authors · 2025-11-16
> > **Thank you! (part 2)**
> >
> > > **Continue A to W2**:
> > >
> > > (3) For the connection between the theoretical framework and the experimental work., we have already provided a pointer in Section 3.1, stating: "More in-distribution (ID) / out-of-distribution (OOD) evaluation details and representative examples are provided in Appendix C."
> > >
> > > To be specific, there are some 1-hop facts: (Paris, CapitalOf, France), (France, LocatedIn, Europe), (Peru, HasNaturalFeature, Andes Mountains) ......
> > >
> > > For non-CoT training ($P(y|x)$): the model memorizes direct 2-hop facts like "(Paris, CapitalOfCountryLocatedIn, Europe)". While this enables it to predict ID tasks "(Berlin, CapitalOfCountryLocatedIn, Europe)", it lacks the reasoning capability to solve OOD tasks such as "(Lima, CapitalOfCountryWithNaturalFeature, Andes Mountains)".
> > >
> > > For CoT training ($P(y|C,x),P(C|x)$): the model learns "(Paris, CapitalOfCountryLocatedIn, Europe)" by combining "(Paris, CapitalOf, France)" and "(France, LocatedIn, Europe)". Therefore, it has the compositional generalization to predict "(Lima, CapitalOfCountryWithNaturalFeature, Andes Mountains)".
> >
> > > **W3**: The robustness analysis feels somewhat detached from the preceding sections. Furthermore, the findings presented in Figure 3 lack a control group for comparison, and the observed trends are intuitively self-evident, which collectively diminishes the insightfulness of this investigation.
> >
> > > **A**: That is a good question! We appreciate the opportunity to clarify this point.
> > >
> > > (1) For robustness analysis, in Section 2.2, we have a robustness discussion of training with CoT (Remark 2, Theorem 3). This corresponds to an increase in the upper bounds of the generalization error for both ID and OOD cases, and the larger the noise ratio $\xi$, the higher
> > the error bounds. Also in Section 3.4, we show that CoT training still enables systematic generalization even with data containing noisy steps. Additionally, we analyze the relationship between noisy samples and those with prediction errors, with further details on the effect of first-hop noise provided in Appendix D.7.
> > >
> > > (2) For the insightfulness, **we are glad that you have noted that "This paper provides both theoretical analysis and experimental results to support their claims and a mechanism analysis is also completed"**, and Reviewer L9CC supports that **"To my knowledge these results are novel"**. Our analysis reveals that the key mechanism behind CoT training
> > lies in its ability to foster compositional generalization: **models learn to systematically combine
> > simpler, previously acquired skills** to tackle novel and more complex problems.

---

> ### Author Response · Authors · 2025-11-16
> **Thank you! (part 3)**
>
> > **W4**: The related work section discusses latent reasoning, yet I fail to see which part of this paper actually relates to it. If the proposed CoT method is indeed performed in a latent space, the authors should explicitly clarify this point in method part. In that case, I would reconsider my first weakness and the overall contribution of this work.
>
> > **A**: You have perfectly grasped the part of core results we aimed to illustrate! (Section 5 in the revised version)
> >
> > (1) Our structural analysis (in Section 3.3) leads a **discussion on the key limitation** of transformer models in OOD generalization.
> >
> > For the circuit with two-hop facts $(e_1,r_1,e_2)\oplus(e_2,r_2,e_3)\rightarrow (e_1,r_1,r_2,e_3)$.
> >
> > ● Training with CoT: the bridge entity ($e_2$) can be extracted from the middle layer hidden states $E(\text{layer index},r_2)$, for ID $\text{layer index}= 3$, and for OOD, $\text{layer index}= 5$ (OOD is more challenging than ID).
> >
> > ● Training without CoT: The generalization circuit fails to emerge in OOD settings, aligning with the performance drop in Figure 1 (left). In ID settings, $\text{layer index} \geq 5$, higher than that in Training with CoT.
> >
> > **Intuitively, smaller $\text{layer index}$ implies that more layers remain available for processing the second hop, potentially leading to better performance**. We also demonstrate that a two-layer transformer is sufficient to learn two-hop compositional circuits from CoT training in Appendix D.3.
> >
> > Only when the intermediate result $e_2$ is resolved can the final result $e_3$ be correctly derived, **traditional transformer cannot process subtasks in parallel**, which is the imitations of traditional transformer architectures.
> >
> > (2) How training with CoT mitigates transformer limits in OOD scenarios.  **The generalization circuit we discovered essentially corresponds to the reuse of the model's weights (effective depth increases)**. That is why we discuss the latent reasoning in the related work.
> >
> > (3) For "**accomplishing multi-step reasoning directly within a single computational process**": To investigate this, we evaluate models trained with CoT—using twice the number of layers with weight reuse—**under a modified input format: “$e_1, r_1, r_2, \text{[mask]}$ ”. This setup is designed to emulate multiple forward passes typically required in next-token prediction.** By examining the intermediate hidden states at the $\text{[mask]}$ position (from the last layer), we observe that the model remains capable of correctly predicting $e_3$. **That is what you said "performed in a latent space" and "within a single computational process"**.
> >
> > We also see a good example of "Loop-transformer Reasoning" [1] works after our initial submission. Our insights further support the validity of this research direction.
> >
> > [1] Scaling Latent Reasoning via Looped Language Models. ByteDance Seed. arXiv:2510.25741.
>
> Thank you once more for your review. We would be pleased to address any additional questions or concerns you might have. Wishing you every success in your research and personal endeavors.

---

> ### Author Response · Authors · 2025-11-26
> **Looking forward to your reply!**
>
> Dear Reviewer ECGg,
>
> We hope this message finds you well.
>
> We have carefully addressed the your comments and made substantial revisions to our manuscript, as detailed in the responses and the revised PDF file. We have also summarized the key changes and enriched content to facilitate your review.
>
> Given that the review timeline is approaching its deadline, **we kindly request your feedback on the revised submission at your earliest convenience**. Your insights and comments are crucial for further improving the quality of our work, and we greatly value the opportunity for continued discussion.
>
> Thank you very much for your time and effort. Please do not hesitate to let us know if there are any additional clarifications or further details needed.
>
> **Sending our best wishes for health, happiness, and all good things in life**！
>
> The Authors

---

> > ### Comment · Reviewer_ECGg · 2025-11-26
> >
> > Thanks to the authors for their answers. My concerns are all addressed.

---

> > > ### Author Response · Authors · 2025-11-26
> > > **Sincerely thank you again!**
> > >
> > > Dear Reviewer ECGg,
> > >
> > > We are deeply grateful for your positive feedback and for the time and effort you have invested in your thoughtful review. Your valuable insights have significantly enriched our work. Thank you once again for your invaluable contribution.
> > >
> > > Warmest regards,
> > >
> > > The Authors

---

### Official Review · Reviewer_pBps · 2025-10-30

**Soundness:** 2
**Presentation:** 3
**Contribution:** 3
**Rating:** 2
**Confidence:** 4

**Summary:**

The paper
- theoretically proves that the information-theoretic generalization bounds through distributional divergence can be decomposed into ID and OOD components.
- empirically proves that CoT training accelerates convergence and enhances generalization from ID to both ID and OOD scenarios.
- also shows interesting findings that CoT-trained models resolve intermediate results at shallower layers and learns how to think by CoT training.

**Strengths:**

The papers empirical finding is interesting that CoT training can help OOD generalization. The paper also combines theory with empirical findings, which is a good try.

**Weaknesses:**

I think the biggest issue is that you cannot directly assume "under suffcient training, the ID generalization error approaches zero" without any justifications. This correlates to what architecture and tasks you are considering. For example, the model can't perform well on the train dataset on the star-graph problem by next-token prediction even if you trained the model suffciently[1]. Thus, in my opinion, it is better to have the view from optimization, which might be hard to analyze. Or you can make more assumptions on the model/task.

[1] Bachmann, G., & Nagarajan, V. (2024). The pitfalls of next-token prediction. arXiv preprint arXiv:2403.06963.

I think the work itself is great, but I think this weakness matters a lot to me. I will increase my score depending on your explanation.

**Questions:**

Q1: In your experiments, why you split the dataset into 2 parts where one part is to output thought given prompt and the other part is to output final answer given prompt+thought. I think it is equivalent to do SFT on thought+final answer given the prompt as input. Why do you formalize the dataset like that?

Q2: grammar error at line 474, should be "an architectural"

---

> ### Author Response · Authors · 2025-11-16
> **Thank you! (part 1)**
>
> Dear Reviewer pBps,
>
> Thank you for your time and encouraging recognition of "the work itself is great". Your comments are highly appreciated, and we have addressed each point in the following responses. We also have revised the manuscript and marked the changes in **orange**.
>
> > **W1**: I think the biggest issue is that you cannot directly assume "under suffcient training, the ID generalization error approaches zero" without any justifications. This correlates to what architecture and tasks you are considering. For example, the model can't perform well on the train dataset on the star-graph problem by next-token prediction even if you trained the model suffciently[1]. Thus, **in my opinion, it is better to have the view from optimization, which might be hard to analyze**. Or you can make more assumptions on the model/task.
> >
> > [1] Bachmann, G., & Nagarajan, V. (2024). The pitfalls of next-token prediction. arXiv preprint arXiv:2403.06963.
>
> > **A**: We sincerely thank the reviewer for raising this important point, we appreciate the opportunity to further summarize and clarify.
> >
> > Firstly, **we agree with your "this correlates to what architecture and tasks you are considering"**.
> >
> > (1) However, we wish to **gently clarify a potential point of misunderstanding**.
> >
> > For your comment (directly assume "under suffcient training, the ID generalization error approaches zero" without any justifications). (i) In Section 2.1 (Lines 102-118), **we have clarified that our focus is on "compositional reasoning tasks," where ID and OOD are defined by whether these compositional patterns are seen during training or are novel and unseen**. Representative examples are provided in Appendix C. (ii) In Section 2.2 (Lines 152-161), we have also shown that under sufficient training—with or without CoT—the ID generalization error approaches zero, **consistent with the results in Figure 1 (Left Part)**. Therefore, **we do not directly assume without any justifications as you said**. (We have updated Lines 214-215 to avoid this misunderstanding.)
> >
> > (2) In the example [1] work you provided, "next-token prediction during training (i.e., teacher-forcing)
> can give rise to problematic learning mechanisms that are detrimental to even in-distribution performance" & "identify that the next-token prediction debate must not conflate autoregressive inference with teacher-forcing". **In our setting, we use the next-token prediction via autoregression rather than next-token prediction via teacher-forcing** (please see **A to Q1** for details, we have updated Lines 226-235 and cited the [1] you provided).
> >
> > (3) Your suggestion "have the view from optimization" is highly constructive. According to the traditional statistical learning viewpoint, performance can be defined by the sum of optimization error and generalization error. (updated in Section 5 of the revised version).
> >
> > (i) For generalization, as defined in Section 2, **the expected generalization error  = population risk - empirical risk**. More specifically, on the one hand, Theorems 1 and 2 show that CoT training helps reduce the **expected generalization error**. On the other hand, we can evaluate the **empirical risk** by assessing the model's performance on the validation set (for example, Figure 1). If the generalization error is known, it is possible to estimate the **population risk**. **Therefore, the key aspect is addressing the generalization error**.
> >
> > (ii) For optimization, from an intuitive perspective (Lines 345-352), CoT-trained models learn to systematically compose simpler skills ($P(C|X) , P(Y |X, C)$), whereas those trained without CoT merely match patterns ($P(Y|X)$) seen in the training data. That is, the optimization object for CoT training is $||P(C_{target}|X)-P(C_{predict}|X)||^2+||P(Y_{target}|X,C_{target})-P(Y_{predict}|X,C_{predict})||^2$, which may be finer-grained (if the snowballing errors [1] are within an acceptable range) than optimization object for non-CoT training $||P(Y_{target}|X)-P(Y_{predict}|X)||^2$. We will formally consider this aspect in our future work.
> >
> > (4) For architecture, our structural analysis (in Section 3.3) leads a discussion on the **key limitation of transformer models** in OOD generalization.
> >
> > For the circuit with two-hop facts $(e_1,r_1,e_2)\oplus(e_2,r_2,e_3)\rightarrow (e_1,r_1,r_2,e_3)$.
> >
> > ● Training with CoT: the bridge entity ($e_2$) can be extracted from the middle layer hidden states $E(\text{layer index},r_2)$, for ID $\text{layer index}= 3$, and for OOD, $\text{layer index}= 5$ (OOD is more challenging than ID).
> >
> > ● Training without CoT: The generalization circuit fails to emerge in OOD settings, aligning with the performance drop in Figure 1 (left). In ID settings, $\text{layer index} \geq 5$, higher than that in Training with CoT.

---

> > ### Author Response · Authors · 2025-11-16
> > **Thank you! (part 2)**
> >
> > > **Continue A to W1**: Intuitively, smaller $\text{layer index}$ implies that more layers remain available for processing the second hop, potentially leading to better performance. We also demonstrate that a two-layer transformer is sufficient to learn two-hop compositional circuits from CoT training in Appendix D.3.
> > >
> > > Only when the intermediate result $e_2$ is resolved can the final result $e_3$ be correctly derived, traditional transformer **cannot process subtasks in parallel**, which is the imitations of traditional transformer architectures.
> > >
> > > How training with CoT mitigates transformer limits in OOD scenarios.  **The generalization circuit we discovered essentially corresponds to the reuse of the model's weights (effective depth increases)**. That is why we discuss the latent reasoning in the related work.
> >
> > > **Q1**: In your experiments, why you split the dataset into 2 parts where one part is to output thought given prompt and the other part is to output final answer given prompt+thought. I think it is equivalent to do SFT on thought+final answer given the prompt as input. Why do you formalize the dataset like that?
> >
> > > **A**: That is a good question! We are glad to have the opportunity to clarify.
> > >
> > > **There may be some misunderstandings regarding the CoT training paradigm**. In our Section 3.1 "Training with CoT: $e_1, r_1, r_2 \xrightarrow{\text{predict}} \hat{e}_2 \text{ and } e_1, r_1, r_2, \hat{e}_2 \xrightarrow{\text{predict}} \hat{e}_3$, predict both the bridge entity $e_2$ and the final tail entity $e_3$". Also in Section 3.3 "In the second-hop stage, the **auto-regressive model uses the $\hat{e}_2$ generated in the first-hop stage."  We did not split the dataset into 2 parts (such as "e1, r1, r2" and "e1, r1, r2, e2") as you mentioned, but strictly adhered to the next-token prediction paradigm [1,2]** (that is $x_t=M(x_1,...,x_{t-1})$). This is **not equivalent** to do SFT on thought+final answer given the prompt as input. (We have updated Lines 226-235.)
> > >
> > > [1] Towards Understanding the Universality of Transformers for Next-Token Prediction. ICLR 2025.
> > >
> > > [2] On the Generalization Ability of Next-Token-Prediction Pretraining. ICML 2025.
> >
> > > **Q2**: grammar error at line 474, should be "an architectural"
> >
> > > **A**: We appreciate your thorough and professional review. The grammar error has been corrected in the revised manuscript.
> >
> > Thank you again for your review.  We would be pleased to address any additional questions or concerns you might have. Wishing you every success in your research and personal endeavors.

---

> ### Comment · Reviewer_pBps · 2025-11-20
> **Reviewer's Reply**
>
> For W1, I think I might not express my point of view very clearly. What I want ideally is that you can prove the convergence of the model (namely not directly assume the convergence on math) and you base the further analysis on the convergence point. It is OK that you give empirical justification for it but it would make the paper's result not very strong in math. The reason for why I cited the next token prediction paper is that I want to show the convergence on math is not obvious (in some tasks the model won't directly converge to the global minimum) and there may be no need for you to cite this paper in your work because it may not be that relevant.
>
> For Q1, I think it would be better to write the " Training without/with CoT" part in a "definition" way to make it clearer because I don't grasp the idea in the very beginning, for example 1. what does \hat mean (it means the output generated by the model, different from the ground truth) 2. you define what D_{train} and D_{test} is and 3. you define $l_train=\mathbb[E]_{D_{train}}[...], l_test=\mathbb{E}_{D_{test}}[...]$. I think the manuscript would benefit from it, though this version is ok for me to understand.
>
> You should also to polish the part of "Training Data" because it is really hard for me to go through what you did about training data and testing data and what you really used for the training and testing. I think misunderstandings all come from here and I am a bit uncertain here for my comprehensive judgement.

---

> > ### Author Response · Authors · 2025-11-21
> > **Thank you again! (part 1)**
> >
> > Dear Reviewer pBps,
> >
> > We sincerely appreciate you **taking the time to further clarify your perspective**. Below, we deal with your new comments point by point.
> >
> > > For W1, I think I might not expressive my point of view very clearly. What I want ideally is that you can prove the convergence of the model (namely not directly assume the convergence on math) and you base the further analysis on the convergence point. **It is OK that you give empirical justification** for it but it would make the paper's result not very strong in math. The reason for why I cited the next token prediction paper is that I want to show the convergence on math is not obvious (in some tasks the model won't directly converge to the global minimum) and there may be no need for you to cite this paper in your work because it may not be that relevant.
> >
> > > **A**: Your feedback has helped us better understand your concerns regarding the mathematical treatment of convergence in our work. Thank you again for your thoughtful and constructive input.
> > >
> > > We **fully agree with your professional view** that "the convergence on math is not obvious" (especially in math formulation). As we clarified in **A to W1**, the convergence assumption correlates to what architecture and tasks we are considering (updated in the revised version). **To ground this assumption more firmly, we discuss the empirical part (training loss and data) as follows.**
> > >
> > > According to the traditional statistical learning viewpoint, performance can be defined by the sum of optimization error and generalization error. The expected generalization error  = population risk - empirical risk. More specifically, on the one hand, Theorems 1 and 2 show that CoT training helps reduce the **expected generalization error**. On the other hand, we can evaluate the **empirical risk** by assessing the model's performance on the validation set (for example, Figure 1). If the generalization error is known, it is possible to estimate the **population risk**. Therefore, **we mainly focus on the generalization error part**. Regarding the mathematical proof of convergence, we agree with your observation that this "might be hard to analyze". We plan to look into this more deeply in our future work.
> >
> > > For Q1, **I think it is better to define what you used as training loss** in math for your empirical part because I am not very sure whether I grasp your paper now. From my understanding of your part 2, you are saying that you generate e_3 only based on \hat e_1, r_1, r_2, if we view \hat e_2 as a function of e_1, r_1, r_2? I think it would also be better to write the setup in a "definition" way, for example 1. what does \hat mean (it means the output generated by the model, different from the ground truth) 2. l_train=𝔼_Dtrain[...] l_test=𝔼_Dtest[...] and 3. you define what D_{test} is. I think the manuscript would benefit from it.
> >
> > > **A**: Thank you for raising this important point!
> > >
> > > (1) Let \hat denotes the output generated by the model, which is different from the ground truth.  In training with CoT, the model first generates $\hat{e}_2$ based on $e_1, r_1, r_2$, then it auto-regressively generates $\hat{e}_3$ based on $e_1, r_1, r_2, \hat{e}_2$. **Therefore, "if we view $\hat e_2$ as a function of $e_1, r_1, r_2$", the model generates $\hat{e}_3$ only based on $e_1, r_1, r_2$. This differs from the formulation mentioned in your comment** "generate $e_3$ only based on $\hat e_1, r_1, r_2$" (difference in notation).
> > >
> > > (2) We agree that adding clear definitions will significantly improve the logical clarity of our manuscript. We will revise the setup section:
> > >
> >  > (i) Definition 3 (Training without/with CoT)—— Let $\hat{e}$ denotes the output generated by the model $\mathcal{M}$, which is different from the ground truth. As for two-hop facts ($e_1,r_1,r_2,e_3$).
> > > Thus, the test loss is defined as $\frac{1}{|S^{(2)}_{test}|}||$ $e_3-\mathcal{M}(e_1,r_1,r_2)||^2$,
> > > where |set| denotes the number of samples in the set.
> > >
> > > For training, we consider whether to use CoT annotations:
> > >
> > > (a) Training without CoT, the loss is $\frac{1}{|S^{(2)}_{\text{train}}|}||e_3-\mathcal{M}(e_1,r_1,r_2)||^2$;
> > >
> > > (b) Training with CoT, the loss is $\frac{1}{|S^{(2)}_{\text{train}}|}(||e_3-\mathcal{M}(e_1,r_1,r_2,\hat{e}_2)||^2+||e_2-\mathcal{M}(e_1,r_1,r_2)||^2)$, where $\hat{e}_2=\mathcal{M}(e_1,r_1,r_2)$.

---

> > ### Author Response · Authors · 2025-11-26
> > **Looking forward to your reply (thank you again)!**
> >
> > Dear Reviewer pBps,
> >
> > We hope this message finds you well.
> >
> > We have carefully addressed the your new comments and made substantial revisions to our manuscript, as detailed in the responses and the revised PDF file. We have also summarized the key changes and enriched content to facilitate your review.
> >
> > Given that the review timeline is approaching its deadline, **we kindly request your feedback on the revised submission at your earliest convenience**. Your insights and comments are crucial for further improving the quality of our work, and we greatly value the opportunity for continued discussion.
> >
> > Thank you very much for your time and effort. Please do not hesitate to let us know if there are any additional clarifications or further details needed.
> >
> > **Sending our best wishes for health, happiness, and all good things in life**！
> >
> > The Authors

---

> > > ### Comment · Reviewer_pBps · 2025-11-26
> > > **Reply from the reviewer**
> > >
> > > I thank the author for the detailed explanations. My concerns are all addressed, therefore I would raise my score accordingly.

---

> > > > ### Author Response · Authors · 2025-11-26
> > > > **Sincerely thank you again!**
> > > >
> > > > Dear Reviewer pBps,
> > > >
> > > > Thank you very much for your positive feedback. We truly appreciate your acknowledgment that our responses have successfully addressed your previous questions. Your continued support and positive rating of the paper are immensely encouraging. Thank you once again for your thoughtful review and valuable insights.
> > > >
> > > > Warmest regards,
> > > >
> > > > The Authors

---

> ### Author Response · Authors · 2025-11-21
> **Thank you again! (part 2)**
>
> > **continue A**:
> >
> > (ii) Definition 4 (ID/OOD Evaluation)—— As we stated in Section 3.1, we denote $S$ as the atomic facts and $S^{(2)}$ as the two-hop facts. The training dataset $T$ excludes ID and OOD test sets,
> > that is, $T = S   \cup S_{\text{ID}_{\text{train}}}^{(2)}$.
> >
> > Notice that  $S_{\text{ID}}^{(2)}\cap S_{\text{OOD}}^{(2)}=\emptyset$ and the **relation compositions ($r_i, r_j$)** in $S_{\text{ID}}^{(2)}$ will not appear in $S_{\text{OOD}}^{(2)}$.
> >
> > **The key distinction between ID and OOD data lies in whether their distributions are the same**. We provide a more prominent example with distribution differences here. The training set is composed of $S_{\text{ID}}, S_{\text{OOD}}$ and $S_{\text{ID}_{\text{train}}}^{(2)}$.
> >
> > $S_{\text{ID}}:(\text{Paris}, \text{CapitalOf}, \text{France}),(\text{France}, \text{LocatedIn}, \text{Europe})$,
> >
> > $(\text{Berlin}, \text{CapitalOf}, \text{Germany}),(\text{Germany}, \text{LocatedIn}, \text{Europe})$.
> >
> >-----
> >
> > $S_{\text{OOD}}: (\text{Lima}, \text{CapitalOf}, \text{Peru}),(\text{Peru}, \text{HasNaturalFeature}, \text{Andes Mountains}).
> S_{\text{ID}_{\text{train}}}^{(2)}$,
> >
> > a uniformly random subset of the inferred facts derived from $S_{\text{ID}}:(\text{Paris}, \text{CapitalOfCountryLocatedIn}, \text{Europe})$.
> >
> >-----
> >
> > $S_{\text{ID}_{\text{test}}}^{(2)}$,
> >
> > previously unseen inferred facts derived from $S_{\text{ID}} (\text{ID generalization}):(\text{Berlin}, \text{CapitalOfCountryLocatedIn}, \text{Europe})$.
> >
> >-----
> >
> > $S_{\text{OOD}}^{(2)}$,
> >
> > previously unseen inferred facts derived from $S_{\text{OOD}}$ (OOD generalization):$\(\text{Lima}, \text{CapitalOfCountryWithNaturalFeature}, \text{Andes Mountains})$.
> >
> > Here, **ID test data has the same relation types** "CapitalOfCountryLocatedIn" with training dara. **OOD test data** requires the model to tackle greater knowledge distribution differences. That is, the **new relation types**, such as "HasNaturalFeature".
> >
> > Thus, as our discussed in Theorem 1 (Remark 1), "For ID, the compositional patterns C in the test problems exactly match those encountered during training, since the ID test data comprises instances derived from known compositions in training data. Thus, under sufficient training—with or without CoT—the ID generalization error approaches zero, consistent with the results in Figure 1 (Left Part)."  **We do not directly assume without any justifications, and this correlates to the ID/OOD definitions we are considering.  However, we are deeply grateful for your meticulous and patient review, and we will add those in the revised version**.
>
> > You should also to polish this part (3.1 General Setup) because it is really hard for me to go through what you did about training data and testing data and what you really used for the training and testing. I think **misunderstandings all come** from here and I am a bit uncertain here for my comprehensive judgement.
>
> > **A**: Kindly refer to the above for "what you did about training data and testing data and what you really used for the training and testing". We have already provided a pointer in Section 3.1, stating: "More in-distribution (ID) / out-of-distribution (OOD) evaluation details and representative examples are provided in Appendix C." **We hope this clarifies any misunderstandings**.

---

> ### Author Response · Authors · 2025-11-21
> **Thank you again! (part 3)**
>
> Moreover, **in addition to the theoretical contributions, please allow us to summarize our structural analysis**:
>
> >(1) Our structural analysis (in Section 3.3) leads a discussion on the **key limitation of transformer models in OOD generalization**.
> >
> > For the circuit with two-hop facts $(e_1,r_1,e_2)\oplus(e_2,r_2,e_3)\rightarrow (e_1,r_1,r_2,e_3)$.
> >
> > ● Training with CoT: the bridge entity ($e_2$) can be extracted from the middle layer hidden states $E(\text{layer index},r_2)$, for ID $\text{layer index}= 3$, and for OOD, $\text{layer index}= 5$ (OOD is more challenging than ID).
> >
> > ● Training without CoT: The generalization circuit fails to emerge in OOD settings, aligning with the performance drop in Figure 1 (left). In ID settings, $\text{layer index} \geq 5$, higher than that in Training with CoT.
> >
> > **Intuitively, smaller $\text{layer index}$ imply that more layers remain available for processing the second hop, potentially leading to better performance**. We also demonstrate that a two-layer transformer is sufficient to learn two-hop compositional circuits from CoT training in Appendix D.3.
> >
> > Only when the intermediate result $e_2$ is resolved can the final result $e_3$ be correctly derived, traditional transformer **cannot process subtasks in parallel**, which is the imitations of traditional transformer architectures.
> >
> > ---
> >
> > (2) How training with CoT mitigates transformer limits in OOD scenarios.  **The generalization circuit we discovered essentially corresponds to the reuse of the model's weights (effective depth increases)**. That is why we discuss the latent reasoning in the related work.
> >
> > (3) Regarding the OOD generalization capability for compositional tasks, what we would truly like to see is whether the model can accomplish multi-step reasoning directly within a single computational process. To investigate this, we evaluate models trained with CoT—using twice the number of layers with weight reuse—under a modified input format: “$e_1, r_1, r_2, \text{[mask]}$”. This setup is designed to emulate multiple forward passes typically required in next-token prediction. By examining the intermediate hidden states at the [mask] position (from the last layer), we observe that the model remains capable of correctly predicting $e_3$.
> >
> > We also see a good example of "Loop-transformer Reasoning" [1] works after our initial submission. **Our insights further support the validity of this research direction**.
> >
> > [1] Scaling Latent Reasoning via Looped Language Models. ByteDance Seed. arXiv:2510.25741.
> We respectfully ask the reviewer to reconsider our contribution in light of these clarifications. Best wishes to you!
>
> Thank you once more for your time and willingness to increase the score. Please let us know if you have any further questions. **We respectfully ask the reviewer to reconsider our contribution in light of these clarifications. Best wishes to you!**

---

### Official Review · Reviewer_L9CC · 2025-11-05

**Soundness:** 3
**Presentation:** 3
**Contribution:** 2
**Rating:** 6
**Confidence:** 3

**Summary:**

This paper provides a theoretical explanation, using generalization bounds to explain why training with CoT data is helpful. The idea is that when the model needs to predict y given x, then the probability can be decomposed as $P(y|x) = \sum_C P(y|X,C)P(C|x)$. If the model has seen the intermediate contexts $C$ then generalization is possible, which does not happen for training without CoT. Experimentally, the authors use a synthetic set-up; the model is trained on 1,2-hop facts 1)without CoT (trained on 1-hop and some 2-hop, evaluated on 2-hop and 2) with CoT (trained on 1-hop and 2-hop together with the intermediate 1-hop facts to perform the 2-hop facts). Their experiments show improved performance and generalization when the model is trained with CoT.

**Strengths:**

1. The authors provide theoretical results to explain why it can be the case that CoT training leads to better generalization. To my knowledge these results are novel.
2. The paper is clearly written.
3. The experiments also consider noisy data to represent more closely real training data.

**Weaknesses:**

1. The experimental set-up is not particularly novel. I think it is quite equivalent to other experimental set-ups with synthetic datasets like for example addition. In addition, for example, models are trained with for example one digit addition and some two digit and it has observed that when CoT is also provided the models generalize and perform better (see [1] for example).

**Questions:**

1. Did the authors tried to perform CoT prompting to the models that were trained without CoT data and see if the performance was improved? The models know each individual component, CoT training basically teaches how to combine these components.

2. Could the authors provide na example from their experimental set-up that matches the theoretical results and shows how CoT training can help predict the correct answer while non-CoT training is unable to do so?

Typo: line 173 the sum should be over C.

---

> ### Author Response · Authors · 2025-11-16
> **Thank you!**
>
> Dear Reviewer L9CC,
>
> Thank you so much for your kind acknowledgment that, to your knowledge, “these results are novel.” We truly appreciate your positive feedback on the theoretical and experimental contributions of our work. Your feedback is invaluable in improving our paper. Please find our detailed response below.
>
> >  **W1**: The experimental set-up is not particularly novel. I think it is quite equivalent to other experimental set-ups with synthetic datasets like for example addition. In addition, for example, models are trained with for example one digit addition and some two digit and it has observed that when CoT is also provided the models generalize and perform better (see [1] for example).
>
> > **A**: That is a good question! We appreciate the opportunity to clarify this point.
> >
> > (1) Role of the Experiment: As stated in Section 3, "this section further illustrates compositional generalization, with our experiments serving as empirical support for the aforementioned theoretical claims": CoT training helps models learn how to think—by enabling compositional reasoning—not just what to think, by simply providing correct answers. We are also pleased that the Reviewer ECGg noted, "**The use of a controlled synthetic task allows for a crisp demonstration of the core principle—compositional generalization**."
> >
> > (2) Connection to addition tasks: Our analysis reveals that the key mechanism behind CoT training
> lies in its ability to foster compositional generalization: models learn to **systematically combine
> simpler, previously acquired skills** to tackle novel and more complex problems. Therefore, (i) **at a high level, we acknowledge the similarities between them**, both single-digit addition tasks and 1-hop facts can be viewed as simple skills, while two-digit addition (or 2-hop facts) represent more complex tasks. (ii) **However, the compositional rule in digit addition tasks is singular**, whereas our task, as discussed in Section 3.1, considers **diverse** relation compositions ($r_i, r_j$).
> >
> > (3) In addition, to better address your comment and ensure we accurately contextualize our contributions relative to existing literature, **could you kindly provide the full citation for reference [1]**? We appreciate your time and constructive comments.
>
> > **Q1**: Did the authors tried to perform CoT prompting to the models that were trained without CoT data and see if the performance was improved? The models know each individual component, CoT training basically teaches how to combine these components.
>
> > **A**: You have perfectly grasped the core results we aimed to illustrate! **We strongly concur with your observation** that "CoT training basically teaches how to combine these components," which **aligns directly with our emphasis** in Section 1 that "compositional generalization is fundamental: models systematically combine simpler learned skills during CoT training to address novel and more complex problems."
> >
> > Besides, **we have conducted precisely this investigation**, as detailed in Section 3.2 and Appendix D.4. Models trained exclusively on atomic facts (**trained without CoT**) exhibit **limited reasoning** capability for compositional facts. However, when we artificially (**perform CoT prompting**) split a compositional fact into atomic facts for testing, the model is also **able to generalize** effectively.
>
> > **Q2**: Could the authors provide na example from their experimental set-up that matches the theoretical results and shows how CoT training can help predict the correct answer while non-CoT training is unable to do so?
>
> > **A**: We sincerely thank the reviewer for raising this important point. **We have already provided a pointer in Section 3.1**, stating: "More in-distribution (ID) / out-of-distribution (OOD) evaluation details and representative examples are provided in Appendix C."
> >
> > To be specific, there are some 1-hop facts: (Paris, CapitalOf, France), (France, LocatedIn, Europe), (Peru, HasNaturalFeature, Andes Mountains) ......
> >
> > For non-CoT training ($P(y|x)$): the model memorizes direct 2-hop facts like "(Paris, CapitalOfCountryLocatedIn, Europe)". While this enables it to predict ID tasks "(Berlin, CapitalOfCountryLocatedIn, Europe)", it lacks the reasoning capability to solve OOD tasks such as "(Lima, CapitalOfCountryWithNaturalFeature, Andes Mountains)".
> >
> > For CoT training ($P(y|C,x),P(C|x)$): the model learns "(Paris, CapitalOfCountryLocatedIn, Europe)" by combining "(Paris, CapitalOf, France)" and "(France, LocatedIn, Europe)". Therefore, it has the compositional generalization to predict "(Lima, CapitalOfCountryWithNaturalFeature, Andes Mountains)".
>
> > **Q3**: Typo: line 173 the sum should be over C.
>
> > **A**: We appreciate your thorough and professional review. The typo has been corrected in the revised manuscript.
>
> We would be pleased to address any additional questions or concerns you might have. Wishing you every success in your research and personal endeavors.

---

> ### Author Response · Authors · 2025-11-26
> **Looking forward to your reply!**
>
> Dear Reviewer L9CC,
>
> We hope this message finds you well.
>
> We have carefully addressed the your comments and made substantial revisions to our manuscript, as detailed in the responses and the revised PDF file. We have also summarized the key changes and enriched content to facilitate your review.
>
> Given that the review timeline is approaching its deadline, **we kindly request your feedback on the revised submission at your earliest convenience**. Your insights and comments are crucial for further improving the quality of our work, and we greatly value the opportunity for continued discussion.
>
> Thank you very much for your time and effort. Please do not hesitate to let us know if there are any additional clarifications or further details needed.
>
> **Sending our best wishes for health, happiness, and all good things in life**！
>
> The Authors

---

### Author Response · Authors · 2025-11-29
**General Response (part 1)**

Dear Area Chair, Senior Area Chair and Program Chair,

We would like to express our deepest gratitude for your generous time and patience.

While our point-by-point rebuttal has directed the reviewers to the exact locations, we **wish to provide a concise summary of these responses for your convenience**, given that they are cardinal to the evaluation of our work's contribution and completeness.

We are grateful to all reviewers for their detailed comments and their **``finally positive scores``** (25 Nov 2025 (AOE), **``long before the ICLR-reported "bug", can be checked in the Revisions``**) of our work! We are encouraged to see that reviewers find:

> ``Reviewer L9CC`` (score 6->6): The authors provide **theoretical results to explain** why it can be the case that CoT training leads to better generalization. To my knowledge **these results are novel**. ``The paper is clearly written``.
>
> ---
>
> ``Reviewer pBps`` (score 2->6): I think the work itself is great...... The papers **empirical finding is interesting** that CoT training can help OOD generalization. **The paper also combines theory with empirical findings, which is a good try**.
>
> (Reviewer pBps's **``final reply``**, : "I thank the author for the detailed explanations. **``My concerns are all addressed``**, therefore I would raise my score accordingly." https://openreview.net/revisions?id=uhJOhIBs1R)
>
> ---
>
> ``Reviewer ECGg`` (score 4->6): This paper provides both **theoretical analysis and experimental results to support** their claims and a mechanism analysis is also completed, which makes the paper **comprehensive**...... The use of a controlled synthetic task allows for a **crisp demonstration of the core principle—compositional generalization**.
>
> (Reviewer ECGg's **``final reply``**: "Thanks to the authors for their answers. **``My concerns are all addressed``**." https://openreview.net/revisions?id=cUXziBe5Ta)

The reviewers' primary concerns focused on several key areas, which we have addressed with detailed responses and new analyses. The revisions are marked with **``orange``** text in the PDF. We summarize these points below.

> **Point1**: The main contribution of this work.
>
> Our analysis reveals that the key mechanism behind CoT training lies in its ability to foster compositional generalization: **models learn to systematically combine simpler, previously acquired skills to tackle novel and more complex problems**.
>
> ---
>
> **(1**, for ``Reviewer pBps, Reviewer ECGg``) **Theoretically**, as defined in Section 2, **the expected generalization error  = population risk - empirical risk**. More specifically, on the one hand, Theorems 1/2 show that CoT training helps reduce the **expected generalization error**. On the other hand, we can evaluate the **empirical risk** by assessing the model's performance on the validation set (for example, Figure 1). If the generalization error is known, it is possible to estimate the **population risk**. Therefore, we mainly focus on the generalization error part.
>
> **(2**, for ``Reviewer pBps, Reviewer ECGg``) **Empirically**, as we stated in Section 3, **our experiments serving as empirical support for the aforementioned theoretical claims**.  Figures 1 and 2 support Theorems 1 and 2 (ID&OOD discussion, compositional circuit), Figure 3 supports Theorem 3 (noise discussion).
>
> **(3**, for ``Reviewer L9CC, Reviewer pBps, Reviewer ECGg``) For **the example from their experimental set-up that matches the theoretical results**, we have already provided a pointer in Section 3.1, stating: "More in-distribution (ID) / out-of-distribution (OOD) evaluation details and representative examples are provided in Appendix C."
>
> To be specific, there are some 1-hop facts: (Paris, CapitalOf, France), (France, LocatedIn, Europe), (Peru, HasNaturalFeature, Andes Mountains) ......
>
> For non-CoT training ($P(y|x)$): the model memorizes direct 2-hop facts like "(Paris, CapitalOfCountryLocatedIn, Europe)". While this enables it to predict ID tasks "(Berlin, CapitalOfCountryLocatedIn, Europe)", it lacks the reasoning capability to solve OOD tasks such as "(Lima, CapitalOfCountryWithNaturalFeature, Andes Mountains)".
>
> For CoT training ($P(y|C,x),P(C|x)$): the model learns "(Paris, CapitalOfCountryLocatedIn, Europe)" by combining "(Paris, CapitalOf, France)" and "(France, LocatedIn, Europe)". Therefore, it has the compositional generalization to predict "(Lima, CapitalOfCountryWithNaturalFeature, Andes Mountains)".

---

> ### Author Response · Authors · 2025-11-29
> **General Response (part 2)**
>
> > **Continue to Point 1**:
> >
> > **(4**, for ``Reviewer ECGg``) **Structural analysis** (latent reasoning):
> >
> > (i) Our structural analysis (in Section 3.3) leads a discussion on the **key limitation of transformer models in OOD generalization**.
> >
> > For the circuit with two-hop facts $(e_1,r_1,e_2)\oplus(e_2,r_2,e_3)\rightarrow (e_1,r_1,r_2,e_3)$.
> >
> > ● Training with CoT: the bridge entity ($e_2$) can be extracted from the middle layer hidden states $E(\text{layer index},r_2)$, for ID $\text{layer index}= 3$, and for OOD, $\text{layer index}= 5$ (OOD is more challenging than ID).
> >
> > ● Training without CoT: The generalization circuit fails to emerge in OOD settings, aligning with the performance drop in Figure 1 (left). In ID settings, $\text{layer index} \geq 5$, higher than that in Training with CoT.
> >
> > **Intuitively, smaller $\text{layer index}$ imply that more layers remain available for processing the second hop, potentially leading to better performance**. We also demonstrate that a two-layer transformer is sufficient to learn two-hop compositional circuits from CoT training in Appendix D.3.
> >
> > Only when the intermediate result $e_2$ is resolved can the final result $e_3$ be correctly derived, traditional transformer **cannot process subtasks in parallel**, which is the imitations of traditional transformer architectures.
> >
> > ---
> >
> > (ii) How training with CoT mitigates transformer limits in OOD scenarios.  **The generalization circuit we discovered essentially corresponds to the reuse of the model's weights (effective depth increases)**. That is why we discuss the latent reasoning in the related work.
> >
> > (iii) Regarding the OOD generalization capability for compositional tasks, what we would truly like to see is whether the model can accomplish multi-step reasoning directly within a single computational process. To investigate this, we evaluate models trained with CoT—using twice the number of layers with weight reuse—under a modified input format: “$e_1, r_1, r_2, \text{[mask]}$”. This setup is designed to emulate multiple forward passes typically required in next-token prediction. By examining the intermediate hidden states at the [mask] position (from the last layer), we observe that the model remains capable of correctly predicting $e_3$.
> >
> > We also see a good example of "Loop-transformer Reasoning" [1] works after our initial submission. **Our insights further support the validity of this research direction**.
> >
> > [1] Scaling Latent Reasoning via Looped Language Models. ByteDance Seed. arXiv:2510.25741.
>
> > **Point 2**: The training data and testing data we used.
> >
> > Regarding the CoT training paradigm, we use the next-token prediction via autoregression. And **we have refined the wording to prevent any potential misunderstandings** (from ``Reviewer pBps, Reviewer ECGg``):
> >
> > Let $\hat{e}$ denotes the output generated by the model $\mathcal{M}$, which is different from the ground truth. Taking two-hop facts $(e_1,r_1,r_2,e_3)$ as an example, the test loss is defined as
> > $E_{S^{(2)}_{\text{test}}}[\mathcal{L}(e_3,\mathcal{M}(e_1,r_1,r_2))]$, where $\mathcal{L}$ denotes the loss function.
> >
> > For training, (i) Training without CoT, the loss is $E_{S^{(2)}_{\text{train}}}[\mathcal{L}(e_3,\mathcal{M}(e_1,r_1,r_2))]$;
> >
> > (ii) Training with CoT, the loss is $E_{S^{(2)}_{\text{train}}}[\mathcal{L}(e_3,\mathcal{M}(e_1,r_1,r_2,\hat{e}_2))+\mathcal{L}(e_2,\mathcal{M}(e_1,r_1,r_2))]$.
>
> > **Point 3**: Additional experiments to support our findings.
> >
> > (1, for ``Reviewer L9CC``) Perform CoT prompting to the models that were trained without CoT data. **We have conducted precisely this investigation**, as detailed in Section 3.2 and Appendix D.4. Models trained exclusively on atomic facts (**trained without CoT**) exhibit **limited reasoning** capability for compositional facts. However, when we artificially (**perform CoT prompting**) split a compositional fact into atomic facts for testing, the model is also **able to generalize** effectively.
> >
> > (2, for ``Reviewer ECGg``) Control group for comparison for the findings presented in Figure 3. For robustness analysis, in Section 2.2, we have a robustness discussion of training with CoT (Remark 2, Theorem 3). This corresponds to an increase in the upper bounds of the generalization error for both ID and OOD cases, and the larger the noise ratio $\xi$, the higher the error bounds. Also in Section 3.4, we show that CoT training still enables systematic generalization even with data containing noisy steps. Additionally, we analyze the relationship between noisy samples and those with prediction errors, with further details on the effect of first-hop noise provided in Appendix D.7.
>
> Thank you again for your consideration. **We sincerely hope that honest authors receive fair recognition**, and we hope that our general response will be helpful for you.
>
> **Sending our best wishes for health, happiness, and all good things in life!**
>
> The Authors

---

### Meta-Review · Area_Chair_HP2s · 2026-01-14

**Summary:**

This paper provides a theoretical and empirical analysis of how CoT enables compositional generalization in LLMs. The authors derive information-theoretic bounds decomposing generalization error into in-distribution (ID) and out-of-distribution (OOD) components. They demonstrate that CoT allows models to combine simpler learned skills to solve complex OOD tasks (something that is lacking in absence of CoT).

Despite initial mixed ratings, the paper has strong merits that justify acceptance, particularly after a successful rebuttal phase where the authors addressed key theoretical and experimental concerns. I agree with the reviewers that the contributions of the paper are interesting and warrant acceptance. I recommend acceptance.

**Reviewer Concerns:**

Reviewer pBps raised concern about assumption that the ID generalization error approaching zero under sufficient training. The authors clarified that their bounds apply to the specific class of compositional reasoning tasks defined in the paper, distinct from the specific counter-examples cited by the reviewer.

Reviewer ECGg argued that the proposed CoT paradigm was possibly just learning two separate single-step tasks rather than genuine compositional reasoning. This concern was sufficiently addressed in the rebuttal process. Overall Reviewer ECGg acknowledged that their concerns were addressed.

Reviewer L9CC maintained positive score during the whole review process.

**Reviewer Scores:**

The paper initially received scores of 6, 2, and 4. However, following the rebuttal, the scores improved to 6, 6, and 6.

Reviewer ECGg (Score Change: 4 to 6), Reviewer pBps (Score Change: 2 to 6) and Reviewer L9CC maintained their score.

---

### Decision · Program_Chairs · 2026-01-26

Accept (Poster)